# Self-Supervised Category-Level Articulated Object Pose Estimation with Part-Level SE(3) Equivariance

**Xueyi Liu**[1,7]**, Ji Zhang**[2]**, Ruizhen Hu**[3]**, Haibin Huang**[4]**, He Wang**[5]**, Li Yi**[1,6,7]
[1]Tsinghua University  [2]Fudan University  [3]Shenzhen University  [4]Kuaishou Technology
[5]Peking University  [6]Shanghai Artificial Intelligence Laboratory  [7]Shanghai Qi Zhi Institute

## Abstract

Category-level articulated object pose estimation aims to estimate a hierarchy of articulation-aware object poses of an unseen articulated object from a known category. To reduce the heavy annotations needed for supervised learning methods, we present a novel self-supervised strategy that solves this problem without any human labels. Our key idea is to factorize canonical shapes and articulated object poses from input articulated shapes through part-level equivariant shape analysis. Specifically, we first introduce the concept of part-level SE(3) equivariance and devise a network to learn features of such property. Then, through a carefully designed fine-grained pose-shape disentanglement strategy, we expect that canonical spaces to support pose estimation could be induced automatically. Thus, we could further predict articulated object poses as per-part rigid transformations describing how parts transform from their canonical part spaces to the camera space. Extensive experiments demonstrate the effectiveness of our method on both complete and partial point clouds from synthetic and real articulated object datasets. The project page with code and more information can be found at: equi-articulated-pose.github.io.

## 1 Introduction

Articulated object pose estimation is a crucial and fundamental computer vision problem with a wide range of applications in robotics, human-object interaction, and augmented reality Katz & Brock (2008); Mu et al. (2021); Labbé et al. (2021); Jiang et al. (2022); Goyal et al. (2022); Li et al. (2020b). Different from 6D pose estimation for rigid objects Tremblay et al. (2018); Xiang et al. (2017); Sundermeyer et al. (2018); Wang et al. (2019a), articulated object pose estimation requires a hierarchical pose understanding on both the object-level and part-level Li et al. (2020a). This problem has been long studied on the instance level where an exact CAD model is required to understand the pose of a specific instance. Recently, there is a trend in estimating category-level object pose such that the algorithm can generalize to novel instances. Despite such merits, supervised category-level approaches always assume rich annotations that are extremely expensive to acquire Li et al. (2020a); Chi & Song (2021); Liu et al. (2022a). To get rid of such restrictions, we tackle this problem under a self-supervised setting instead.

Given a collection of unsegmented articulated objects in various articulation states with different object poses, our goal is to design a network that can acquire a category-level articulated object pose understanding in a self-supervised manner without any human labels such as pose annotations, segmentation labels, or reference frames for pose definition.

The self-supervised category-level articulated object pose estimation problem is highly ill-posed since it requires the knowledge of object structure and per-part poses, which are usually entangled with part shapes. Very few previous works try to solve such a problem or even similar ones. The most related attempt is the work of Li et al. (2021). It tackles the unsupervised category-level pose estimation problem but just for rigid objects. They leverages SE(3) equivariant shape analysis to disentangle the global object pose and shape information so that a category-aligned canonical object space can emerge. This way the category-level object poses could be automatically learned by predicting a transformation from the canonical space to the camera space. Going beyond rigid objects, estimating

articulated object poses demands more than just global pose and shape disentanglement. It requires a more fine-grained disentanglement of part shape, object structure such as part adjacency relationship, joint states, part poses, and so on.

To achieve such fine-grained disentanglement, we propose to leverage part-level SE(3) equivariant shape analysis. Especially, we introduce the concept of part-level SE(3) equivariant features to equip equivariance with a spatial support. The part-level SE(3) equivariant feature of a local region should only change as its parent part transforms but should not be influenced by the transformation of other parts. This is in contrast to the object-level SE(3) equivariant feature for a local region, which is influenced by both the region's parent part and other parts. To densely extract part-level SE(3) equivariant features from an articulated shape, we propose a novel pose-aware equivariant point convolution operator. Based on such features, we are able to achieve a fine-grained disentanglement which learns three types of information from input shapes: 1) *Canonical part shapes*, which are invariant to input pose or articulation changes and are category-aligned to provide a consistent reference frame for part poses; 2) *Object structure*, which is also invariant to input pose or articulation changes and contains structural information about the part adjacency relationships, part transformation order, and joint parameters such as pivot points; 3) *Articulated object pose*, which is composed of a series of estimated transformations. Such transformations include per-part rigid transformations which assembles canonical part shapes into a canonical object shape, per-part articulated transformation which articulates the canonical object shape to match the input articulation state, and a base part rigid transformation transforming the articulated canonical object to the camera space. To allow such disentanglement, we guide the network learning through a self-supervised part-by-part shape reconstruction task that combines the disentangled information to recover the input shapes.

With the above self-supervised disentanglement strategy, our method demonstrates the possibility of estimating articulated object poses in a self-supervised way for the first time. Extensive experiments prove its effectiveness on both complete point clouds and partial point clouds from various categories covering both synthetic and real datasets. On the Part-Mobility Dataset Wang et al. (2019b), our method without the need for any human annotations can already outperform the iterative pose estimation strategy with ground-truth segmentation masks on both complete and partial settings by a large margin, *e.g.* reduce the rotation estimation error by around 30 degrees on complete shapes and by 40 degrees on partial shapes. Besides, our method can perform on par with to or even better than supervised methods like NPCS Li et al. (2020a). For instance, we can achieve an average of $7.9°$ rotation estimation error on complete shapes, comparable to NPCS's $5.8°$ error. We can even outperform NPCS on some specific categories such as partial Eyeglasses. Finally, we prove the effectiveness of our part-level SE(3) equivariance design and the fine-grained disentanglement strategy in the ablation study. Our main contributions are summarized as follows:

- To our best knowledge, we are the first that tackles the self-supervised articulated object pose estimation problem.
- We design a pose-aware equivariant point convolution operator to learn part-level SE(3)-equivariant features.
- We propose a self-supervised framework to achieve the disentanglement of canonical shape, object structure, and articulated object poses.

## 2 RELATED WORKS

**Unsupervised Part Decomposition for 3D Objects.** Decomposing an observed 3D object shape into parts in an unsupervised manner is a recent interest in shape representation learning. Previous works always tend to adopt a generative shape reconstruction task to self-supervise the shape decomposition. They often choose to represent parts via learnable primitive shapes Tulsiani et al. (2017); Kawana et al. (2020); Yang & Chen (2021); Paschalidou et al. (2021); Deng et al. (2020); Zhu et al. (2020); Chen et al. (2020) or non-primitive-based implicit field representation Chen et al. (2019); Kawana et al. (2021). Shape alignment is a common assumption of such methods to achieve consistent decomposition across different shapes.

**Articulated Object Pose Estimation.** Pose estimation for articulated objects aims to acquire a fine-grained understanding of target articulated objects from both the object level and the part level. The prior work Li et al. (2020a) proposes to estimate object orientations, joint parameters, and

per-part poses in a fully-supervised setting. They define Articulation-aware Normalized Coordinate Space Hierarchy (ANCSH), composed of the canonical object space and a set of canonical part spaces, as a consistent representation for articulated objects to support pose estimation. In this work, we also want to estimate a hierarchy of articulation-aware object poses but in a totally unsupervised setting. Instead of hand-crafting normalized coordinate spaces, we wish to let them be automatically induced during learning.

**SE(3) Equivariant Networks.** Recently, there is a trend of pursuing SE(3)-equivariant and invariant features through network design Weiler et al. (2018); Thomas et al. (2018); Fuchs et al. (2020); Zhao et al. (2020); Chen et al. (2021). Equivariance is achieved by designing kernels Thomas et al. (2018); Fuchs et al. (2020) or designing feature convolution strategies Chen et al. (2021); Zhao et al. (2020). In this work, we design our part-level SE(3) equivariant feature network based on Equivariant Point Network Chen et al. (2021) for articulated object pose estimation. Common SE(3) equivariant feature of a local region would be affected by both its parent part's and other parts' rigid transformations. By contrast, its part-level SE(3) equivariant feature would only be affected by its parent part.

## 3 METHOD

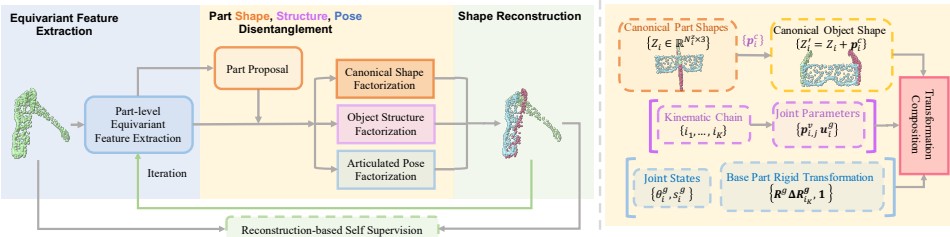

Figure 1: Overview of the proposed self-supervised articulated object pose estimation strategy. The method takes a complete or partial point cloud of an articulated object as input, factorizes canonical shapes, object structure, and the articulated object pose from it. The network is trained by a shape reconstruction task. **Left:** A high-level abstraction of our pipeline. **Right:** An illustrate of decomposed information for shape reconstruction. Green lines ($\leftarrow$) denote the iterative pose estimation process.

We present our method for self-supervised category-level articulated object pose estimation. We first propose to learn part-level SE(3) equivariant features through a novel pose-aware equivariant point convolution module (sec. 3.1). Based on such features, we then design a disentanglement strategy to factorize an arbitrarily posed 3D point cloud into three types of information. Such information includes a canonical shape with category-aligned pose and articulation state, the object structure describing the part adjacency and joints, as well as the articulated object pose (sec. 3.2). We find part-level SE(3) equivariant features are key to achieve the factorization above. Further, we adopt a part-by-part shape reconstruction task that combines the factorized information for shape reconstruction to self-supervise the factorization (sec. 3.3). Our method assumes a category-level setting where input shapes have the same kinematic chain. For notations frequently used in the following text, $N$, $C$, and $K$ denote the number of points, feature dimension, and the number of parts per shape respectively.

### 3.1 PART-LEVEL SE(3)-EQUIVARIANT NETWORK

We first elaborate on our part-level SE(3) equivariant network. The network $\phi(\cdot)$ operates on a point cloud $X = \{\mathbf{x}_i | 1 \leq i \leq N\}$ with per-point pose and outputs part-level SE(3) equivariant features for all points $F = \{F_i = \phi(X)[i] | 1 \leq i \leq N\}$. Here the pose of a point refer to the pose of that point's parent part. We introduce the concept of part-level equivariant feature to differentiate from object-level equivariant features in Chen et al. (2021), where the per-point feature changes equivariantly with the global transformation applied to the object. Part-level equivariant feature $F_i$ of each point $x_i$ changes equivariantly with the rigid transformation applied to its parent part, but remains invariant to transformations of other parts. We develop our network based on the Equivariant Point Network (EPN) Chen et al. (2021) with a novel pose-aware equivariant point convolution module to support part-level equivariance. In the following text, we would briefly review EPN, and then continue with our pose-aware equivariant point convolution.

**Equivariant Point Network.** EPN takes a point cloud $X$ containing $N$ points and a rotation group $G$ with $|G|$ elements as input and extracts $C$-dimensional per-point per-rotation features, forming a feature matrix $F \in \mathbb{R}^{N \times C \times |G|}$. $F$ is rotational and translational equivariant to a specific rigid transformation group $G_A$ induced by $G$. The rotational equivariant transformation for each rotation element $g \in G$ in the feature domain is a corresponding matrix permutation of $F$ along the last dimension. The translational equivariance achieved by EPN is essentially translational invariance. Simply using relative point coordinates for convolution allows $F$ to remain the same while translating the input point cloud.

**Pose-aware Equivariant Point Convolution.** For part-level SE(3) equivariant features, we design a pose-aware point convolution strategy that operates on a point cloud with per-point poses. While conducting convolution within a group of points, our core idea is to align point poses to the pose of the group center. Since we use the pose of a point to refer to the pose of its parent part, such alignment could cancel out the influence of the varying articulation states on the geometric description of each point. Intuitively speaking, if a point comes from the same part as the group center, information will just get aggregated as a normal convolution. While a point comes from a different part from the group center, pose alignment will canonicalize the articulation state so that the convolution outcome remains the same regardless of the articulation state change. Our pose-aware convolution strategy allows aggregating context information from different parts but avoids feature changing as the articulation changes. Equipping EPN with this strategy, we are able to achieve part-level equivariance since the feature of each point only changes as its parent part transforms but remains invariant to the transformation of other parts. We then formally define our convolution operator. Taking a point cloud $X$ and the per-point pose $P = \{P_i | 1 \leq i \leq N\}$ as input, our convolution operator for the point $x_i$'s feature at the rotation element $g$ is as follows:

$$(\mathcal{F} * h_1)(x_i, g) = \sum_{x_j \in \mathcal{N}_{x_i}} \mathcal{F}(x_j, g\mathbf{R}_i\mathbf{R}_j^{-1}) h_1(g(x_i - P_i P_j^{-1} x_j)), \tag{1}$$

where $\mathcal{F}(x_i, g)$ is an input function, $h_1(x_i, g)$ is a kernel function, $\mathcal{N}_{x_i}$ is the set of points in the neighbourhood of $x_i$, $P_i$ and $P_j$ denote the input pose of point $x_i$ and point $x_j$ respectively, $\mathbf{R}_i$ and $\mathbf{R}_j$ is their rotation components. We prove that using the above convolution within EPN leads to part-level equivariance in the Appendix A.2. We highlight that we adopt an iterative pose estimation strategy (see Appendix A.3 for details) for the per-point poses and rotations in Eq. 1, which are initialized to be identity in the first iteration.

### 3.2 Part Shape, Structure, and Pose Disentanglement

To obtain a fine-grained understanding of an articulated object, we disentangle three types of information from the input: 1) Canonical shape; 2) Object structure; 3) Articulated object pose. To be more specific, we first use the designed part-level SE(3)-equivariant network to extract per-point features from an input shape. We then leverage a self-supervised slot-attention module to group the featured points, forming a set of featured parts for the disentanglement. We predict a canonical shape for each part to induce the category-level canonical part spaces required by part pose definition. Then we disentangle structure and pose-related information that gradually transform canonical part shapes to the observed shape. First, we predict *part-assembling parameters* to transform each canonical part shape to form the canonical object shape. After that, the *kinematic chain*, *joint parameters* and *joint states* are predicted to articulate the canonical object shape into the observed articulation state. Finally, a *base part rigid transformation* is predicted to further transform the resulting articulated object to the observed shape in the camera space. We will elaborate details of the above designs in the following text.

**Part Proposal.** The part proposal module groups $N$ points in the input shape $X$ into $K$ parts for per-part equivariant features extraction. It learns an invariant grouping function that maps $X$ together with a point feature matrix $F$ to a point-part association matrix $\mathbf{W} \in \mathbb{R}^{N \times K}$. Specifically, we adopt an attention-pooling operation for the per-point invariant feature together with a slot attention module Locatello et al. (2020) for the grouping purpose. Based on the proposed parts, we can group points in the input shape $X$ into $K$ point clouds $\{X_i | 1 \leq i \leq K\}$ and compute the per-part equivariant feature $\{F_i | 1 \leq i \leq K\}$.

**Shape: Canonical Part Shape Reconstruction.** With per-part equivariant features, we aim to predict a canonical shape for each part which should be aligned within a certain category so that the category-level part pose can be defined. The canonical shape for each part should be invariant to every parts' rigid transformations. Thus, we adopt an SE(3)-invariant canonical shape reconstruction module constructed based on an SO(3)-PointNet module as utilized in Li et al. (2021). The reconstruction module converts per-part equivariant features $F_i$ into per-part invariant features through attention pooling first and then predicts an SE(3)-invariant shape $Z_i$ for each part.

**Structure: Kinematic Chain Prediction.** In addition to the canonical shape of each part, we also need to understand the kinematic chain of a shape. The kinematic chain defines how different parts are connected and the order they get transformed when a cascaded transformation happens. , *i.e.* from chain leaves to the chain root. To estimate the kinematic chain for a given shape, we first construct an adjacency confidence graph from object parts and then extract its maximum spanning tree consisting of the set of confident adjacency edges. We set the part with the largest degree in the graph to be the root of the tree, which will also serve as the base part of the object. The transformation order is further predicted as the inverse DFS visiting order of the tree. Notice the kinematic chain should not be affected by the articulated input pose, we therefore leverage per-part SE(3)-invariant features for estimation.

**Structure: Joint Parameters Prediction.** For each pair of adjacent parts , we will then infer their joint parameters, including an invariant pivot point $\mathbf{p}_{i,j}^v$ and a joint axis orientation hypothesis $\mathbf{u}_i^g$ for each rotation element $g \in G_g$. For pivot points, we treat them as invariant properties and still adopt an invariant shape reconstruction module for prediction. Specifically, we predict the pivot point $\mathbf{p}_{i,j}^v$ between every two adjacent parts $i, j$ from their equivariant feature $(F_i, F_j)$ using an invariant shape reconstruction module Li et al. (2021). For joint axis orientations, we regress an axis orientation hypothesis $\mathbf{u}_i^g$ for part $i$ corresponding to each rotation group element $g \in G_g$ from its equivariant feature $F_i$.

**Pose: Part-assembling Parameters Prediction.** Part-assembling parameters transform the predicted canonical part shapes to assemble a canonical object shape. As parameters connecting invariant canonical shapes, they should be invariant to every parts' rigid transformations as well. Here, we simply predict a translation vector $\mathbf{p}_i^c \in \mathbb{R}^3$ for each part $i$. We predict them through invariant shape reconstruction modules from per-part equivariant feature $\{F_i | 1 \leq i \leq K\}$. We can then assemble predicted canonical part shapes together to form the canonical object shape: $Z = \{Z_i + \mathbf{p}_i^c | 1 \leq i \leq K\}$.

**Pose: Joint States Prediction.** Joint states describe the articulation state of an object. For each part $i$, we predict a joint state hypothesis for each rotation element $g \in G$ from its equivariant feature $F_i$, *i.e.* a rotation angle $\theta_i^g$ for a revolute part or a translation scalar $s_i^g$ for a prismatic part. We can therefore articulate the canonical object shape based on the predicted kinematic chain and joint states with the base part fixed, so as to match the object articulation from the input observation.

**Pose: Base Part Rigid Transformation.** The base part rigid transformation needs to transform the articulated canonical object shape to the camera space. Since we have previously predicted joint states hypotheses for all rotation element $g$, we will also need multiple base transformation hypotheses correspondingly. We simplify the base part transformation to be a rotation, which proves to be effective in practice. A straightforward way is to use the rotation matrix corresponding to each rotation element $g$ as the base transformation hypothesis. We follow this idea but also predict an additional residual rotation as a refinement. By transforming the articulated shape of the canonical object shape via the predicted base part rigid transformation, we can align the resulting shape with the observed input object.

**Articulated Object Pose.** With the above predicted quantities, we can calculate per-rotation articulated object pose hypotheses for an input articulated object $X$, including three parts: 1) translation $\mathbf{p}_i^c$ of each part $i$ which assembles category-aligned canonical parts into a canonical object; 2) per-rotation articulated transformation of the canonical object based upon the predicted kinematic chain, joint parameters and per-rotation joint states; 3) per-rotation base part rigid transformation which transforms the articulated canonical object into the camera space. The rigid transformation hypothesis

for each part $i$ corresponding to each rotation element $g \in G$ is denoted as $P_i^g = (\mathbf{R}_i^g, \mathbf{t}_i^g)$. We treat them as part pose hypotheses.

### 3.3 Shape Reconstruction-based Self-supervised Task

Based on the reconstructed canonical part shapes and predicted per-rotation part pose hypotheses, we can get per-rotation shape reconstruction for each part $i$: $\{Y_i^g = \mathbf{R}_i^g Z_i + \mathbf{t}_i^g | g \in G\}$. A part-by-part reconstruction task is adopted to self-supervise the network. Besides, we add a regularization term for each predicted joint so that the joint indeed connects two parts.

**Shape Reconstruction-based Self-supervised Loss.** The per-rotation shape reconstruction for the whole object can be calculated by concatenating all part reconstructions: $Y^g = \{Y_i^g | 1 \le i \le K\}$. We then adopt a min-of-N loss between the input observation $X$ and the reconstructed posed point clouds:

$$\mathcal{L}_{rec} = \min_{g \in G} d(X, Y^g), \tag{2}$$

where $d : \mathbb{R}^{N_X \times 3} \times \mathbb{R}^{N_Y \times 3} \to \mathbb{R}$ denotes the distance function between two point clouds and could be unidirectional or bidirectional Chamfer Distance as an example.

**Regularization for Joint Prediction.** Predicted joints should connect adjacent parts and support natural articulations. However, just supervising joint parameters from the reconstruction loss is not sufficient for the needs above. Therefore, we devise a point-based joint constraint term for each predicted joint $(\mathbf{u}_i^{g_0}, \mathbf{p}_{i,j}^v)$, where $g_0 = \arg\min_{g \in G} d(X, Y^g)$ (Eq. 2). Specifically, given the predicted pivot point $\mathbf{p}_{i,j}^v$ and joint orientation $\mathbf{u}_i^{g_0}$, we independently randomly sample a set of points from the joint by shifting the pivot point $\mathbf{p}_{i,j}^v$: $P_{i,j}^v = \{\mathbf{p}_{i,j}^{v,k} | 0 \le k \le K^v\}$. The joint regularization loss term is as follows:

$$\mathcal{L}_{reg} = \sum_{(i,j) \in \mathcal{E}_{\mathcal{T}}} d(P_{i,j}^v, Z_i^2) + d(P_{i,j}^v, Z_j^2) + d(P_{i,j}^v, Z_i^1) + d(P_{i,j}^v, Z_j^1),$$

where $Z_i^1$ and $Z_i^2$ are shapes of the part $i$ in the canonical object space before and after its articulated transformation, $\mathcal{E}_{\mathcal{T}}$ is the set of adjacent parts, $d(X_1, X_2)$ is the unidirectional Chamfer Distance function from point cloud $X_1$ to $X_2$.

Our final self-supervised shape reconstruction loss is a linear combination of the above two loss terms: $\mathcal{L} = \mathcal{L}_{rec} + \lambda \mathcal{L}_{reg}$, where $\lambda$ is a hyper-parameter.

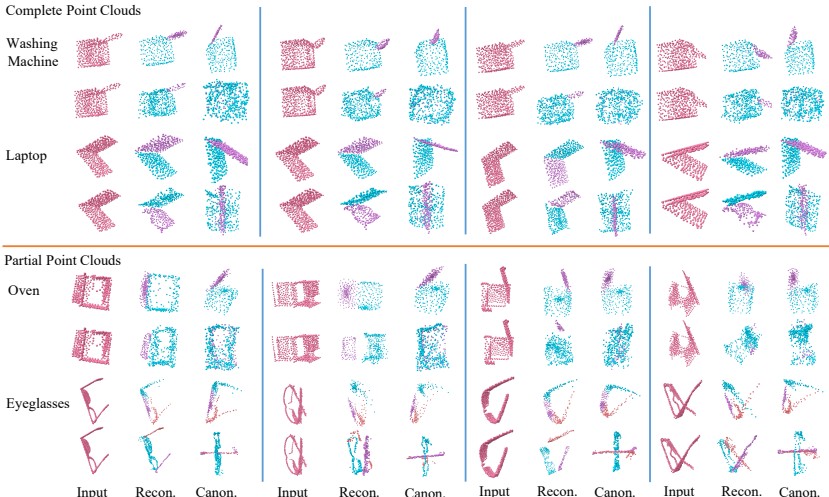

Figure 2: Visualization for qualitative evaluation. For every two lines, the first line draws the results of our method, and the second line draws those of NPCS. Every three shapes from the left side to the right side are the input point cloud (**Input**), reconstruction (**Recon.**), and the reconstructed canonical object shape (**Canon.**). **We do not assume input shape alignment but align them here when drawing just for a better view.** Please zoom in for details.

## 4 EXPERIMENTS

We evaluate our method on the category-level articulated object pose estimation task (sec. 4.2) to demonstrate its effectiveness. Besides, we also test its performance on two side tasks that can be completed by our network at the same time, namely part segmentation (sec. 4.3), and shape reconstruction (sec. 4.4).

### 4.1 DATASETS

Following previous literature Li et al. (2020a), we choose seven categories from three datasets for evaluation on both complete shapes and rendered partial point clouds: 1) Four categories from the Part-Mobility Wang et al. (2019b) dataset, namely Oven, Washing Machine, Laptop (denoted as Laptop (S)), and Eyeglasses with revolute parts. 2) One category, Drawer with prismatic parts, from SAPIEN dataset Thomas et al. (2011). 3) Two categories from a real dataset HOI4D Liu et al. (2022b), namely Safe and Laptop (denoted as Laptop (R)) with revolute parts. Please refer to the Appendix B.1 for data preparation details.

### 4.2 CATEGORY-LEVEL ARTICULATED OBJECT POSE ESTIMATION

**Metrics.** Following Li et al. (2020a), we use the following metrics to evaluate our method: 1) Part-based pose-related metrics, namely per-part rotation error $R_{err}(^{\circ})$ and per-part translation error $T_{err}$, both in the form of mean and median values; 2) Joint parameters, namely joint axis orientation errors $\theta_{err}(^{\circ})$ in degrees and joint position error $d_{err}$, both in the form of mean values. Please refer to the Appendix B.8 for details of our evaluation strategy.

**Baselines.** Since there is no previous works that have exactly the same setting with ours, we choose NPCS Li et al. (2020a), a **supervised** pose estimation method for articulated objects, and ICP, a traditional pose estimation approach, as our baseline methods. To apply them on our articulated objects with arbitrary global poses, we make the following modifications: 1) We change the backbone of NPCS to EPN Chen et al. (2021) (denoted as "NPCS-EPN") and add supervision on its discrete rotation mode selection process to make it work on our shapes with arbitrary global pose variations. We do observe that the NPCS without EPN will fail to get reasonable results on our data (see Appendix B.5 for details). Beyond part poses, we also add a joint prediction branch for joint parameters estimation. 2) We equip ICP with ground-truth segmentation labels (denoted as "Oracle ICP") and register each part individually for part pose estimation. Notice that Oracle ICP cannot estimate joint parameters.

**Experimental Results.** Table 1 presents the experimental results of our method and baseline methods on complete point clouds. We defer the results on partial point clouds to Table 7 in the Appendix B.4. We can make the following observations: 1) As a self-supervised strategy, our average and per-category performance are comparable to that of the supervised baseline NPCS-EPN. We can even sometimes outperform NPCS-EPN such as the joint axis orientation estimation on Safe. 2) Without any human label available during training, our method can outperform the Oracle ICP with ground-truth segmentation labels by a large margin in all categories. As a further discussion, the poor performance of Oracle ICP may be caused by the part-symmetry related problem. It would add ambiguity on part poses especially when we treat each part individually for estimation. Please refer to Appendix C for more discussions. For a qualitative evaluation and comparison, we visualize the input objects, reconstructions, and the predicted canonical object shapes by our method and NPCS in Figure 2. Our method is able to reconstruct category-level aligned canonical shapes, which serve as good support for estimating category-level articulated object poses.

### 4.3 PART SEGMENTATION

**Evaluation Metric and Baselines.** The metric used for this task is Segmentation IoU (MIoU). We choose three position-based segmentation strategies, namely BAE-Net Chen et al. (2019), NSD Kawana et al. (2020), BSP-Net Chen et al. (2020) and one motion-based segmentation method ICP ICP as our baselines for this task. For BAE-NEt and BSP-Net, we generate data in their implicit

Table 1: Comparison between the part pose estimation performance of different methods on all categories. "R" denotes rotation errors with the value format "Mean $R_{err}$/Median $R_{err}$". "T" denotes translation errors with the value format "Mean $T_{err}$/Median $T_{err}$". "J" denotes joint parameters estimation results with the value format "Mean $\theta_{err}$/Mean $d_{err}$". "Avg." refers to "Average Value". **Since Oracle ICP could not predict joint parameters**, we only present joint parameter prediction results of our method and supervised NPCS-EPN. For all metrics, the smaller, the better. Best values and **Bold**, while second best ones are shown in *blue*.

| | Method | Oven | Washing Machine | Eyeglasses | Laptop (S) | Safe | Laptop (R) | Drawer | Avg. |
|---|---|---|---|---|---|---|---|---|---|
| R | NPCS-EPN (supervised) | **5.47/4.45**, *7.35/7.30* | **4.76/4.07**, 6.66/5.41 | **2.75/2.44**, **9.34/7.64**, **7.93/6.74** | **6.72**/*6.08*, *15.96/13.91* | **1.75/1.59**, **2.67/2.50** | *8.20/7.12*, *5.13/4.72* | **1.52/1.31**, **2.01/1.81**, **2.15/1.81**, **1.14/0.94** | **5.38/4.70** |
| | Oracle ICP | 46.46/38.56, 47.11/43.41 | 55.12/50.42, 52.38/51.57 | 34.41/23.25, 34.58/25.82, 35.71/25.12 | 43.26/42.02, 44.04/43.64 | 52.80/56.02, 53.04/52.13 | 42.50/43.06, 42.06/39.25 | 50.15/47.14, 50.12/47.15, 50.11/47.15, 50.07/46.41 | 46.11/42.48 |
| | Ours | *7.74/7.35*, **4.07/3.97** | *7.49/7.37*, 19.27/19.19 | *8.16/8.21*, *12.29/10.89*, *12.53/9.88* | *7.34*/**5.16**, **10.41/9.34** | *9.03/9.09*, *13.83/13.59* | **5.71/3.61**, **3.64/2.84** | *3.18/2.73*, *3.18/2.73*, *3.18/2.71*, *3.18/2.71* | *7.90/7.14* |
| T | NPCS-EPN (supervised) | **0.029/0.029**, 0.020/0.019 | **0.021/0.018**, **0.016**/*0.015* | **0.025/0.025**, **0.022/0.020**, **0.027/0.024** | **0.040**/0.019, **0.027/0.023** | **0.005/0.005**, **0.010/0.009** | **0.014/0.011**, **0.023/0.021** | **0.035**/*0.033*, **0.039**/*0.033*, **0.025/0.016**, **0.013/0.011** | **0.023/0.019** |
| | Oracle ICP | 0.091/*0.041*, 0.070/*0.030* | 0.126/*0.028*, *0.032*/**0.013** | 0.092/0.097, 0.188/0.197, 0.185/0.193 | 0.071/0.037, 0.120/*0.030* | 0.072/*0.036*, 0.060/*0.017* | 0.123/0.122, 0.120/0.123 | *0.053*/**0.029**, *0.054*/**0.027**, *0.050/0.028*, *0.052/0.031* | 0.092/0.063 |
| | Ours | *0.054*/0.052, *0.067*/0.046 | *0.082*/0.083, 0.042/0.034 | *0.054/0.039*, *0.086/0.088*, *0.070/0.055* | **0.040**/*0.037*, *0.046*/0.042 | *0.066*/0.069, *0.037*/0.035 | *0.021/0.019*, *0.027/0.026* | 0.096/0.096, 0.097/0.092, 0.108/0.105, 0.109/0.100 | *0.065/0.060* |
| J | NPCS-EPN (supervised) | **5.04/0.076** | **5.66/0.078** | **7.42**/0.090, **7.42**/0.101 | **5.74**/0.129 | 14.15/0.063 | **8.53/0.084** | **20.18**/- | **9.27/0.089** |
| | Ours | 20.30/0.089 | 28.40/0.118 | 17.75/**0.045**, 17.75/0.129 | 30.31/**0.122** | **4.36/0.031** | 17.17/0.169 | 38.86/- | 21.86/0.100 |

representation using the data generation method described in IM-NET Chen & Zhang (2019). We improve the evaluation strategy for NSD and BSP-Net considering the global pose variation of our data (see Appendix B.3 for details).

**Experimental Results.** In table 2, we present experimental results of our method and baselines on complete point clouds. Results on partial data are deferred to Table 8 in the Appendix B.4. Our method can consistently outperform such four part segmentation methods in all categories. BSP-Net, BAE-Net, and NSD assume input data alignment and highly rely on position information for segmentation. However, such segmentation cues may not be well preserved in our data with arbitrary global pose variations. By contrast, part motions can serve as a more consistent cue for segmenting articulated objects than positions. We hypothesize that ICP's poor registration performance on some categories such as Eyeglasses further lead to its low segmentation IoUs.

## 4.4 SHAPE RECONSTRUCTION

**Evaluation Metric and Baselines.** We choose to use Chamfer L1 as our evaluation metric for shape reconstruction. To demonstrate the superiority of part-by-part reconstruction for articulated objects over the whole shape reconstruction, we choose EPN which treats them as rigid objects for reconstruction as the baseline.

**Experimental Results.** As shown in Table 3, our method can consistently outperform the EPN-based whole shape reconstruction. We suppose part-by-part reconstruction where only simple parts should be recovered makes the reconstruction an easier problem for networks than recovering the whole shape.

Table 2: Comparison between the part segmentation performance of different methods on all categories. Metric used for this task is Segmentation MIoU, calculated on 4096 points for each shape. Values presented in the table are scaled by 100. Larger values indicate better performance. "*" denote cases where the network fails by segmenting input shapes into single parts.

| | Oven | Washing Machine | Eyeglasses | Laptop (S) | Safe | Laptop (R) | Drawer | Avg. |
|---|---|---|---|---|---|---|---|---|
| BAE-Net Chen et al. (2019) | 55.04 | 46.07* | 37.19* | 65.21 | 39.83* | 66.35 | 22.83* | 47.50 |
| NSD Kawana et al. (2020) | 60.59 | 56.43 | 53.31 | 80.88 | 71.30 | 76.86 | 33.61 | 61.85 |
| BSP-Net Chen et al. (2020) | 67.24 | 62.52 | 54.28 | 79.41 | 76.59 | 81.33 | 42.15 | 66.22 |
| Oracle ICP ICP | 75.17 | 72.80 | 49.49 | 56.20 | 66.90 | 59.96 | 45.68 | 60.89 |
| Ours | **76.22** | **73.27** | **62.84** | **82.97** | **80.06** | **86.04** | **51.39** | **73.26** |

Table 3: Comparison between the shape reconstruction performance of different methods on all categories. Metric used in this task is Chamfer L1. The smaller, the better.

| Method | Oven | Washing Machine | Eyeglasses | Laptop (S) | Safe | Laptop (R) | Drawer | Avg. |
|---|---|---|---|---|---|---|---|---|
| EPN Li et al. (2021) | 0.033 | 0.051 | 0.028 | 0.029 | 0.030 | 0.028 | 0.057 | 0.036 |
| Ours | **0.025** | **0.049** | **0.025** | **0.024** | **0.026** | **0.026** | **0.045** | **0.031** |

Table 4: Ablation study w.r.t. the effectiveness of joint regularization for part pose estimation and the design of pose-aware equivariant feature communication (denoted as "Pose."). Reported values are per-category per-part average values. Please refer to the caption of Table 5 for the data format of "Joint".

| Method | Seg. IoU | Mean $R_{err}(°)$ | Median $R_{err}(°)$ | Mean $T_{err}$ | Median $T_{err}$ | Joint | Chamfer L1 |
|---|---|---|---|---|---|---|---|
| No $\mathcal{L}_{reg}$ | 76.40 | 11.74 | 10.87 | 0.070 | 0.065 | - | 0.038 |
| With $\mathcal{L}_{reg}$ | 74.32 | 10.40 | 9.30 | 0.072 | 0.073 | 22.01/0.111 | 0.032 |
| With $\mathcal{L}_{reg}$ (Pose.) | **76.90** | **9.21** | **8.40** | **0.052** | **0.047** | **19.72/0.103** | **0.025** |

## 5    ABLATION STUDY

In this section, we try to ablate some crucial designs in the method to demonstrate their effectiveness, including part-level feature accumulation, pose-aware point convolution, and joint regularization.

**Part-level Feature Accumulation.** We use a grouping module to group points into parts for part-level features in our method. To demonstrate the effectiveness of using part-level features for part shape, structure, pose disentanglement, we ablate part-level features and only use features from the whole shape for part-level properties prediction, similar to those used in Kawana et al. (2021); Chen et al. (2019). Table 5 compares their performance. For each metric, we report its per-category per-part average value. It can be observed that part-level features can help with part-based properties prediction, letting the network achieve better performance on all pose-related metrics.

**Pose-aware Point Convolution.** Our method contains a pose-aware equivariant feature convolution design for part-level SE(3) equivariant feature learning. To demonstrate the superiority of part-level equivariance over common global equivariance, we compare the model's performance when using part-level equivariant features (With $\mathcal{L}_{reg}$ (Pose.)) with the one using global equivariant features (With $\mathcal{L}_{reg}$) in table 4. For each metric, its per-category per-part average value is reported. The network using part-level equivariant features could consistently outperform the one using only global equivariant features on all metrics.

**Joint Regularization.** Besides reconstruction loss, we additionally add a joint regularization term to predict joints that connect two adjacent parts. Beyond acquiring joint-related parameters, joint regularization could improve the pose estimation performance as well, especially for translation prediction, as shown in Table 4.

## 6    CONCLUSION

In this work, we propose a self-supervised strategy for category-level articulated object pose estimation without any annotations. Leveraging part-level SE(3) equivariant features, we propose a part shape, structure, pose disentanglement strategy that successfully accomplish the category-level articulated object pose estimation task. A part-by-part shape reconstruction task is adopted to self-supervise the network learning. Experiments prove the effectiveness of our method and our core ideas. This work can reduce the annotation efforts for solving this tasks and would also promote further thinkings on designing part-level equivariant networks.

Table 5: Ablation study w.r.t. the effectiveness of accumulating part-level features for part-based properties prediction. Reported values are per-category per-part average values on all categories. "Joint" represents joint parameter estimation errors, with the value in the format of "Mean $\theta_{err}$/Mean $d_{err}$".

| Method | Seg. IoU | Mean $R_{err}(°)$ | Median $R_{err}(°)$ | Mean $T_{err}$ | Median $T_{err}$ | Joint | Chamfer L1 |
|---|---|---|---|---|---|---|---|
| Without Parts | 71.68 | 12.85 | 11.52 | 0.068 | 0.060 | 27.97/0.172 | 0.036 |
| With Parts | **76.90** | **9.21** | **8.40** | **0.052** | **0.047** | **19.72/0.103** | **0.029** |

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

The appendix is organized as follows:

- Proofs and further explanations on the method
    - Proof of part-Level equivariant property of the pose-aware point convolution module (sec. A.2).
    - Further explanations on some method components (sec. A.3).
- Experiments
    - Data preparation (sec. B.1).
    - Implementation details (sec. B.2).
    - Implementation details for baselines (sec. B.3).
    - Experiments on partial point clouds (sec. B.4).
    - Additional comparisons and applications (sec. B.5).
    - Robustness to input data noise (sec. B.6).
    - Visualization of part-level equivariant features (sec. B.7).
    - Evaluation strategy for category-level articulated object poses (sec. B.8).
- Discussion on part symmetry (sec. C)

# A  METHOD DETAILS

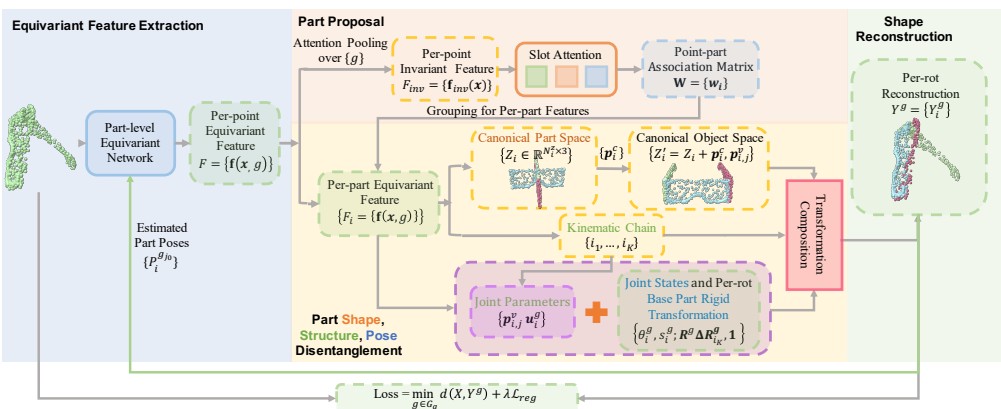

Figure 3: Overview of the proposed self-supervised articulated object pose estimation strategy. The method takes a complete or partial point cloud of an articulated object as input, factorizes canonical shapes, object structure, and the articulated object pose from it. The network is trained by a shape reconstruction task. Part-level SE(3) equivariant features are learned by iterating between part pose estimation and pose-aware equivariant point convolution. Green lines (←) denote procedures for feeding the estimated part poses back to the pose-aware point convolution module.

## A.1  OVERVIEW

We provide an detailed diagram of our self-supervised learning strategy in Figure 3.

## A.2  PROOF OF PART-LEVEL EQUIVARIANT PROPERTY OF THE POSE-AWARE POINT CONVOLUTION MODULE

In this section, we prove the part-level equivariant property of the designed pose-aware point convolution module:

$$(\mathcal{F} * h_1)(x_i, g) = \sum_{P_j^{-1} x_j \in \mathcal{N}^c_{P_i^{-1} x_i}} \mathcal{F}(x_j, g\mathbf{R}_i \mathbf{R}_j^{-1}) h_1(g(x_i - P_i P_j^{-1} x_j)), \tag{3}$$

where $P_i$ and $P_j$ are the (estimated) pose of $x_i$ and $x_j$ from the canonical object space to the camera space respectively, $\mathbf{R}_i$ and $\mathbf{R}_j$ are the (estimated) rotations of point $x_i$ and point $x_j$ from the canonical object space to the camera space respectively, $\mathcal{N}^c_{P_i^{-1} x_i}$ denotes the set of point $x_i$'s neighbours in the canonical object space. Note that the neighbourhood set $\mathcal{N}_{x_i}$ in the Equation 1 represents the neighbourhood of $x_i$ in the camera space with points in which belong to $x_i$'s neighbourhood in the canonical object space, *i.e.* $\mathcal{N}^c_{P_i^{-1} x_i}$. $\mathcal{N}_{x_i}$ would vary as $x_i$'s pose changes, while $\mathcal{N}^c_{P_i^{-1} x_i}$ keeps the same. $\{x_j | x_j \in \mathcal{N}_{x_i}\} = \{x_j | P_j^{-1} x_j \in \mathcal{N}^c_{P_i^{-1} x_i}\}$.

To prove the part-level equivariance of $(\mathcal{F} * h_1)(x_i, g)$, we need to prove 1) $(\mathcal{F} * h_1)(x_i, g)$ is invariant to the rigid transformation of point $x_i$'s each neighbouring point $x_j$; 2) $(\mathcal{F} * h_1)(x_i, g)$ is equivariant to the rigid transformation of $x_i$ itself.

We then prove those properties for the continuous convolution operations, $(\mathcal{F} * h_1)(x_i, g) = \int_{x_j \in \mathbb{R}^3} \mathcal{F}(x_j, g\mathbf{R}_i\mathbf{R}_j^{-1})h_1(g(x_i - P_iP_j^{-1}x_j))$.

**Theorem 1.** *The continuous operation* $(\mathcal{F} * h_1)(x_i, g) = \int_{x_j \in \mathbb{R}^3} \mathcal{F}(x_j, g\mathbf{R}_i\mathbf{R}_j^{-1})h_1(g(x_i - P_iP_j^{-1}x_j))$ *is invariant to each arbitrary rigid transformation* $\Delta P_j = (\Delta\mathbf{R}_j \in SO(3), \Delta\mathbf{t}_j \in \mathbb{R}^3)$ *of* $(x_j \forall x_j \in \mathbb{R}^3, x_j \neq x_i)$ *of* $x$'s *neighbouring point* $x_j$.

*Proof.* To prove the invariance of $(\mathcal{F} * h_1)(x_i, g)$, we need to prove that $\forall x_j \in \mathbb{R}^3, x_j \neq x_i, \forall \Delta P_j \in$ SE(3), $\mathbf{R}'_j = \Delta\mathbf{R}_j\mathbf{R}_j$, we have

$$\Delta P_j(\mathcal{F} * h_1)(x_i, g) = (\mathcal{F} * h_1)(x_i, g).$$

Let $x'_j = \Delta P_j x_j$, $P'_j = \Delta P_j P_j$, then we have,

$$\Delta P_j(\mathcal{F} * h_1)(x_i, g) = \int_{x'_j \in \mathbb{R}^3} \mathcal{F}(x'_j, g\mathbf{R}_i\mathbf{R}_j^{'-1})h_1(g(x_i - P_iP_j^{'-1}x'_j))$$

$$= \int_{x_j \in \mathbb{R}^3} \mathcal{F}(\Delta P_j x_j, g\mathbf{R}_i\mathbf{R}_j^{-1}\Delta\mathbf{R}_j^{-1})h_1(g(x_i - P_iP_j^{-1}\Delta P_j^{-1}\Delta P_j x_j))$$

$$= \int_{x_j \in \mathbb{R}^3} \mathcal{F}(\Delta\mathbf{R}_j x_j, g\mathbf{R}_i\mathbf{R}_j^{-1}\Delta\mathbf{R}_j^{-1})h_1(g(x_i - P_iP_j^{-1}x_j))$$

$$= \int_{x_j \in \mathbb{R}^3} \mathcal{F}(x_j, g\mathbf{R}_i\mathbf{R}_j^{-1})h_1(g(x_i - P_iP_j^{-1}x_j))$$

$$= (\mathcal{F} * h_1)(x_i, g).$$

**Theorem 2.** *The continuous operation* $(\mathcal{F} * h_1)(x_i, g) = \int_{x_j \in \mathbb{R}^3} \mathcal{F}(x_j, g\mathbf{R}_i\mathbf{R}_j^{-1})h_1(g(x_i - P_iP_j^{-1}x_j))$ *is equivariant to the rigid transformation* $\Delta P_i = (\Delta\mathbf{R}_i \in SO(3), \Delta\mathbf{t}_i \in \mathbb{R}^3)$ *of* $x_i$.

*Proof.* To prove that $(\mathcal{F} * h_1)(x_i, g)$ is equivariant to the rigid transformation of $x_i$, we need to prove that $\forall \Delta P_i \in$ SE(3), we have

$$\Delta P_i(\mathcal{F} * h_1)(x_i, g) = (\Delta\mathbf{R}_i\mathcal{F} * h_1)(x_i, g).$$

It can be proved by

$$\Delta P_i(\mathcal{F} * h_1)(x_i, g) = (\mathcal{F} * h_1)(\Delta P_i x_i, g\Delta\mathbf{R}_i)$$

$$= \int_{x_j \in \mathbb{R}^3} \mathcal{F}(x_j, g\Delta\mathbf{R}_i\mathbf{R}_i\mathbf{R}_j^{-1})h_1(g(\Delta P_i x_i - \Delta P_iP_j^{-1}x_j))$$

$$= \int_{x_j \in \mathbb{R}^3} \mathcal{F}(x_j, (g\Delta\mathbf{R}_i)\mathbf{R}_i\mathbf{R}_j^{-1})h_1((g\Delta\mathbf{R}_i)(x_i - P_j^{-1}x_j))$$

$$= (\Delta\mathbf{R}_i\mathcal{F} * h_1)(x_i, g).$$

## A.3 FURTHER EXPLANATIONS ON SOME METHOD COMPONENTS

**Kinematic Chain Prediction.** The kinematic chain is predict as an invariant property from per-part invariant features to describe part articulation transformation order. It is predicted through the following four steps: 1) Predict an adjacency confidence value $c_{i,j}$ for each part pair $(i, j)$; 2) Construct an fully-connected adjacency confidence graph $\mathcal{G} = (\mathcal{V}, \mathcal{E})$ based on predicted confidence values, with all parts as its nodes and predicted confidence values as edge weights; 3) Find a maximum spanning tree from the constructed graph $\mathcal{G}$: $\mathcal{T} = (\mathcal{V}, \mathcal{E}_{\mathcal{T}})$; 4) Calculate the DFS visiting order of $\mathcal{T}$

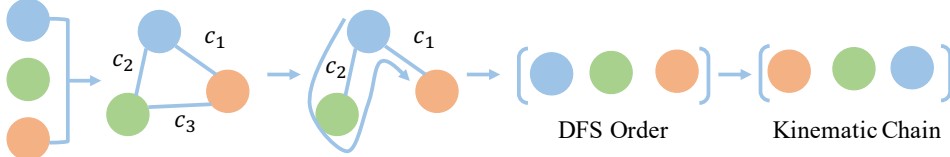

Figure 4: Kinematic chain prediction procedure (an example of the object containing three parts).

and take the inverse visiting order as the predicted kinematic chain. We draw the prediction procedure in Figure 4.

**Invariant/Equivariant Features for Prediction.** Given per-part equivariant feature output from the feature backbone $F_i$, its equivariant feature for equivariant properties prediction is further calculated from an SO(3)-PointNet, *i.e.* $\hat{F}_i = \text{SO(3)-PointNet}(X_i, F_i)$. Its invariant feature for invariant properties prediction is then computed through a max-pooling operation: $F_i^{inv} = \text{Max-Pooling}(\hat{F}_i)$.

**Joint Axis Orientation.** We assume that all joints' axis orientations in one shape are consistent. Thus, in practice, we set the orientation of all joints to the same predicted orientation, *i.e.* $\mathbf{u}_i^g \leftarrow \mathbf{u}_{i_m}^g, \forall (i,j) \in \mathcal{E}_{\mathcal{T}}, \forall g \in G_g$, where $(i_m, j_m)$ is set to the part pair connected to the tree root. Saying $(i,j) \in \mathcal{E}_{\mathcal{T}}$, we mean a directional edge from part $i$ to part $j$. In the node pair $(i,j) \in \mathcal{E}_{\mathcal{T}}$, node $i$ is deeper than node $j$ in the tree $\mathcal{T}$. It indicates that node $i$'s subtree should rotate around the joint $\mathbf{u}_i^g$ passing through the joint between $i$ and $j$.

**Iterative Pose Estimation.** Our pose-aware equivariant point convolution module requires per-point pose as input. Due to our self-supervised setting where input poses are not assumed, we adopt an iterative pose estimation strategy. Through this design, we can improve the quality of part-level equivariant features gradually by feeding back estimated poses in the last iteration to the pose-aware point convolution module in the current iteration. It is because that more accurate input per-point poses would lead to better "part-level" SE(3) equivariant features considering the nature of our pose-aware point convolution. In practice, we set per-point poses to identity values in the first iteration due to the lack of estimated poses, *i.e.* $P_0 = (\mathbf{R}_0, \mathbf{t}_0)$.

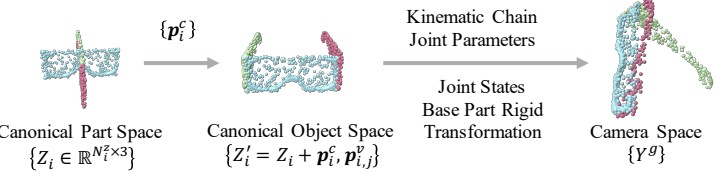

Figure 5: The relationship between our three crucial spaces: the canonical part spaces, the canonical object space, and the camera space.

**Canonical Part Spaces, Canonical Object Space, Camera Space.** For each part, the canonical part space normalizes its pose. For each object, the canonical object space normalizes its object orientation, articulation states. Camera space denotes the observation space. Each part shape in the canonical part space is its canonical part shape. Each object shape with articulation states canonicalized is called its canonical object shape. Canonical *spaces* are category-level concepts, while canonical shapes are instance level concepts. Figure 5 draws the relationship between such three spaces mentioned frequently in our method.

**Partial Point Clouds.** The loss function used in Li et al. (2021) for partial point clouds is the unidirectional Chamfer Distance. Using this function can make the network aware of complete shapes by observing partial point clouds from different viewpoints. Such expectation can be achieved for asymmetric objects if the viewpoint could cover the full SO(3) space. However, we restrict the range of the viewpoint when rendering partial point clouds for articulated objects to make each part visible. Such restriction would result in relatively homogeneous occlusion patterns. Therefore, we choose to use unidirectional Chamfer Distance only for certain categories such as Safe when tested on partial point clouds.

Table 6: Per-category data splitting.

|        | Oven | Washing Machine | Eyeglasses | Laptop (S) | Safe | Laptop (R) | Drawer |
|--------|------|-----------------|------------|------------|------|------------|--------|
| #Total | 32   | 41              | 42         | 82         | 30   | 50         | 30     |
| #Train | 28   | 36              | 37         | 73         | 26   | 44         | 24     |
| #Test  | 4    | 5               | 5          | 9          | 4    | 6          | 6      |

**Equivariant/Invariant Properties of the Designed Modules.** The designed method wishes to use part-level SE(3)-equivariance to reduce the difficulty of factorizing part pose and part shape. Exact part-level equivariant features can make those modules meet our expectations. However, due to the approximate SE(3)-equivariance of the employed feature backbone and the estimated part pose that may not be very accurate, we cannot expect such invariance/equivariance for them. For instance, if we do not consider part kinematic constraints, the part shape reconstruction module and the part-assembling parameters prediction module should be invariant to K rigid transformations in the quotient group $(\text{SE}(3)/G_A)(\text{SE}(3))^{K-1}$ if using global equivariant features, while it should be invariant to the rigid transformation in the quotient group $\text{SE}(3)/G_A$ if using part-level equivariant features given correct part pose estimation. Similarly, the pivot point prediction module should be invariant to two rigid transformations in the quotient group $(\text{SE}(3)/G_A)^2$ if using part-level equivariant features. Part-level equivariance design could reduce the difficulty of a network doing factorization, which may count as a reason for its effectiveness.

## B  EXPERIMENTS

### B.1  DATA PREPARATION

**Data Collection.** We choose seven categories from three different datasets, namely Oven, Washing Machine, Eyeglasses, Laptop (S) with revolute parts from Shape2Motion Wang et al. (2019b), Drawer with prismatic parts from SAPIEN Xiang et al. (2020), Safe and Laptop (R) with revolute parts from HOI4D Liu et al. (2022b).

The first five datasets are selected according to previous works on articulated object pose estimation or part decomposition Li et al. (2020a); Kawana et al. (2021). To further test the effectiveness of our method on objects collected from the real world, we choose two more categories (Safe and Laptop (R)) from a real dataset Liu et al. (2022b).

**Data Splitting.** We split our data according to the per-category data split approach introduced in Li et al. (2020a). Note that not all shapes in a category are used for training/testing. Incomplete shapes or instances whose canonical articulation states are inconsistent with other shapes are excluded from experiments. Per-category train/test splits are listed in Table 6.

**Data Preprocessing.** For each shape, we generate 100 posed shapes in different articulation states.

Then for complete point clouds, we randomly generate 10 rotated samples from each articulated posed object. When generating articulated posed objects, we would add restrictions on valid articulation state ranges. For Oven, Safe, and Washing Machine, the valid degree range of their lids is $[45°, 135°)$. For Eyeglasses, the range of the degree between two legs and the frame is set to $[0°, 81°)$. For Laptop (S) and Laptop (R), the range of the degree between two parts is set to $[9°, 99°)$.

For partial point clouds, we render depth images of complete object instances using the same rendering method described in Li et al. (2021). The difference is that we manually set a viewpoint range for each category to ensure that all parts are visible in the rendered depth images. For each articulated posed shape, we render 10 depth images for it. The dataset will be made public.

**Data samples visualization.** In Figure 6, we provide samples of training and test shapes for some categories for an intuitive understanding w.r.t. intra-category shape variations. Such variations mainly come from part geomtry (*e.g.* Eyeglasses frames, Oven bodies, Laptop) and part size (*e.g.* Washing Machine, Laptop).

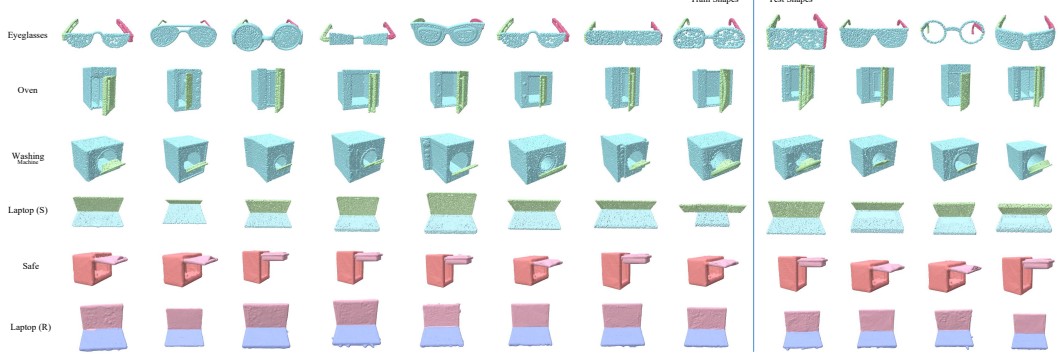

Figure 6: Samples of training and test shapes.

## B.2 IMPLEMENTATION DETAILS

**Architecture.** For point convolution, we use a kernel-rotated version kernel point convolution (KPConv Thomas et al. (2019)) proposed in EPN Chen et al. (2021). The size of the (one) convolution kernel is determined by the number of anchor points and the feature dimension. In our implementation, we use 24 anchor points. Feature dimensions at different convolution blocks are set to 64, 128, and 512 respectively.

**Training Protocol.** In the training stage, the learning rate is set to $0.0001$, which is decayed by 0.7 every 1000 iterations. The model is trained for 10000 steps with batch size 8 on all datasets. We use the self-supervised reconstruction loss to train the network, with the weight for joint regularization $\lambda$ set to 1.0 empirically. We use Adam optimizer with $\beta = (0.9, 0.999), \epsilon = 10^{-8}$.

**Software and Hardware Configurations.** All models are implemented by PyTorch version 1.9.1, torch_cluster version 1.5.1, torch_scatter version 2.0.7, pyrender version 0.1.45, trimesh version 3.2.0, and Python 3.8.8. All the experiments are conducted on a Ubuntu 20.04.3 server with 8 NVIDIA GPUs, 504G RAM, CUDA version 11.4.

## B.3 IMPLEMENTATION DETAILS FOR BASELINES

**NPCS Li et al. (2020a).** The NPCS's original version Li et al. (2020a) trains a network for category-level articulated object pose estimation in a supervised manner. It utilizes a PointNet++ Qi et al. (2017) to regress three kinds of information and a set of pre-defined normalized part coordinate spaces. Then in the evaluation process, the RANSAC algorithm is leveraged to calculate the rigid transformation of each part from its predicted normalized part coordinates to the shape in the camera space. To apply NPCS in our experiments, we make the following two modifications: 1) We change the backbone used in NPCS from PointNet++ to EPN. We further add supervision on its rotation mode selection process for the major rotation matrix prediction as does in Chen et al. (2021). 2) We add a joint axis orientation prediction branch and a pivot point regression branch for joint parameters estimation. Such two prediction branches act on the global shape feature corresponding to the selected rotation mode and predict a residual rotation and a translation for estimation. By applying the major rotation matrix of the selected mode, we could then arrive at the joint axis orientations and pivot points in the camera space.

**Oracle ICP.** To apply ICP on the articulated object pose estimation, we introduce Oracle ICP. Oracle ICP registers each ground-truth part from the template shape to the observed shape iteratively. We randomly select 5 segmented shapes from the train set to register on each test shape. For complete point clouds, we first centralize the part shape from both of the template shape and the observed shape, we then iteratively register the template part shape to the observed part shape under 60 initial hypotheses. For partial point clouds, we iteratively register the template part shape to the observed part shape under 60 initial hypotheses together with 10 initial translation hypotheses. The one that achieves the smallest inlier RMSE value is selected as the registration result. After each registration,

we assign per-point segmentation label as the label of the nearest part. Therefore, we also treat Oracle ICP as one of our segmentation baseline.

**BSP-Net Chen et al. (2020).** BSP-Net reconstructs an input shape using implicit fields as the representation by learning to partition the shape using three levels of representations, from planes to convexes, and further to the concave. Indices of reconstructed covexes are consistent across different shapes in the same category. Thus, we can map from each convex index to a ground-truth segmentation label. The relationship can then help us get segmentations for test shapes. The mapping can then help us segment each test shape by assigning each convex to its corresponding part segment. The intra-category convex partition consistency further provide cross-instance aligned part segmentations. However, due to the global pose variations of our data, such convex index consistency may not be observed when directly applying BSP-Net on our data. Thus, we propose to improve the evaluation process of BSP-Net to mitigate this problem. Specifically, for each test shape, we find a shape from the train set that is the most similar to the current test shape. Then, we directly use its convex-segmentation mapping relationship to get segments for the test shape. The segments are then used to calculate the segmentation IoU for the test shape.

**NSD Kawana et al. (2020).** Neural Star Domain Kawana et al. (2020) decomposes shapes into parameterized primitives. To test its part segmentation performance on our data with arbitrary global pose variations, we adopt the evaluation strategy similar to that used for BSP-Net.

### B.4  EXPERIMENTS ON PARTIAL POINT CLOUDS

In this section, we present the experimental results on rendered partial point clouds of our method and baseline methods.

**Articulated Object Pose Estimation.** In Table 7, we present the part pose estimation and joint parameter prediction results of our method and baseline methods. Compared to the pose estimation results of different models achieved on complete point clouds, our model can sometimes outperform the **supervised** NPCS baseline (using EPN as the backbone), such as better part pose estimations and joint parameter prediction results on the Laptop (S) dataset.

In Figure 8, we draw some samples for a qualitative evaluation. Moreover, we also provide the visualization of all categories for complete point clouds in Figure 7. In the figure, the point cloud distance function used for Safe is unidirectional Chamfer Distance, while that used for others is still bidirectional Chamfer Distance. Using unidirectional Chamfer Distance can relieve the problem of joint regularization on partial point clouds to some extent. It is because that in this way the point cloud completion could be naturally enforced. For instance, reconstruction results for the Safe category are drawn in Figure 8. However, points that are not mapped to any point in the input shape will also affect the point-based joint regularization. For simple shapes, using bidirectional Chamfer Distance could also sometimes make the decoder decode complete part shapes, e.g. reconstructions for Laptop (R). As for the reconstructed reference shape in the canonical object space, better joint predictions would lead to better global shape alignment. For instance, we can observe that the angle between two parts of Laptop (R) and Laptop (S) is relatively consistent across shapes with different articulation states. Joint regularization enforcing the connectivity between two adjacent parts both before and after the articulated transformation in the canonical object space. It could then help make the joint behave like a real joint, based on which we the "lazy" network tends to decode part shapes with consistent orientations. However, there is a degenerated solution where the decoded rotation angle is put near to zero. In that case, the joint regularization term could be satisfied by decoding "twisted" part shapes. Since the decoded angle is near zero, the connectivity between two parts will not be broken when rotating along the decoded joint. Sometimes, decoding angles near to zeros is a local minimum that the optimization process gets stuck in. At that time, the regularization loss term is a large one but the decoded joint parameters are not optimized in the correct direction by the network. The reconstructed shapes in the canonical object space do not have consistent angles, e.g. Washing Machine and Safe drawn in Figure 8.

**Part Segmentation.** In Table 8, we evaluate the segmentation performance of our method. BSP-Net is not compared here since it requires mesh data for pre-processing, which is not compatible with the rendered partial point clouds. Oracle ICP uses real segmentation labels to register each part from

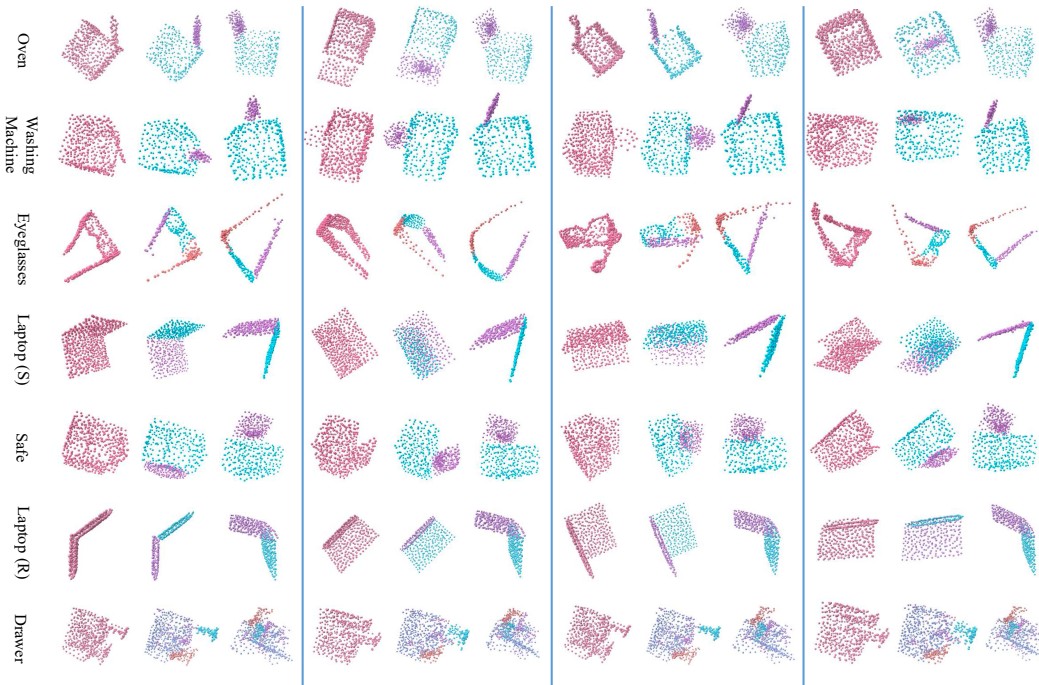

Figure 7: Visualization for experimental results on complete point clouds. Shapes drawn for every three shapes from the left side to the right side are the input point cloud, reconstructions, and the predicted canonical object shape. **We put drawers in an aligned space just for better visualization.** Their global pose may vary when feeding into the network. Please zoom in for details.

the example shape to the observed shape. Despite this, it can still not achieve satisfactory estimation results due to shape occlusions and part-symmetry-related pose ambiguity issues.

**Shape Reconstruction.** In Table 9, we evaluate the shape reconstruction performance of our method. The part-by-part reconstruction strategy used by our method can outperform the EPN-based whole shape reconstruction strategy in most of those categories except for Washing Machine. One possible reason is the poor segmentation performance of our model on shapes in the Washing Machine category.

## B.5 Additional Comparisons and Applications

**Comparison with Other Baselines.** We compare our method with other two baselines that are not discussed in the main body in Table 10.

Firstly, we use KPConv Thomas et al. (2019) as NPCS's feature backbone (denoted as "NPCS-KPConv") and test its performance on our data with arbitrary global pose variation. We can see that NPCS of this version performs terribly compared to our unsupervised method. NPCS estimates part poses by estimating the transformation from estimated NPCS coordinates and the observed shape. It therefore requires invariant NPCS predictions to estimate category-level part poses. However, such prediction consistency may not be easily achieved for input shapes with various global pose variations.

The second one is Oracle EPN, where we assume ground-truth part segmentation labels and use EPN to estimate the pose for each individual part. Despite in such oracle setting, EPN cannot infer joint parameters since it estimates per-part poses individually. Besides, the part symmetry problem will also hinder such strategy from getting good performance to some extent, which will be discussed in the next section C.

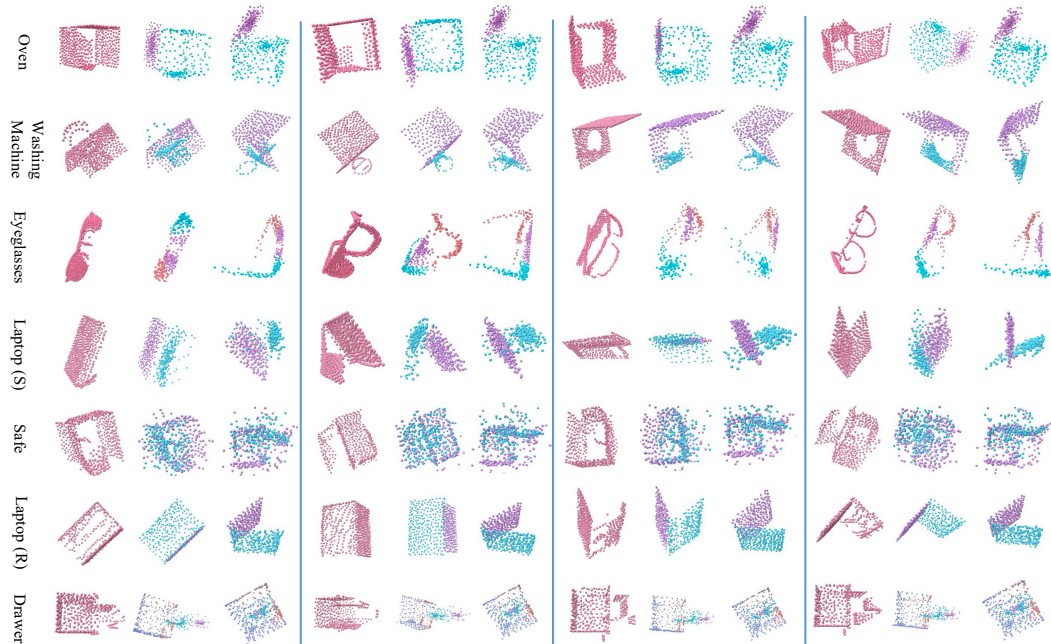

Figure 8: Visualization for experimental results on **partial point clouds**. Shapes drawn for every three shapes from the left side to the right side are the input point cloud, reconstructions, and the predicted canonical object shape. Please zoom in for details.

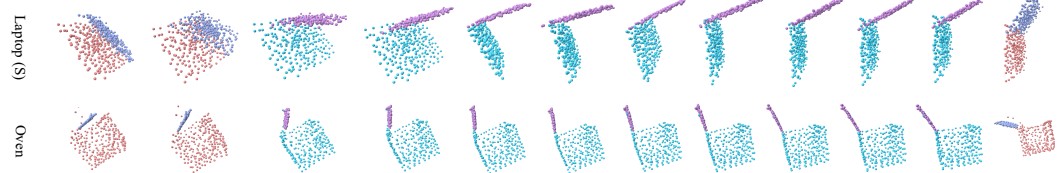

Figure 9: Reconstruction for shapes in different articulation states and manipulations to change their states. Shapes (in blue and orange) drawn on the two sides are manipulated shapes from their nearest reconstructions. Others are reconstructions (in purple and green). Please zoom in for details.

**Shape Reconstruction and Manipulation.** The predicted joints can enable us to manipulate the reconstruction by changing the value of predicted rotation angles. We then arrive at shapes in new sarticulation states different from input shapes. In Figure 9, we draw some examples for Laptop (S) and Oven.

## B.6 ROBUSTNESS TO INPUT DATA NOISE

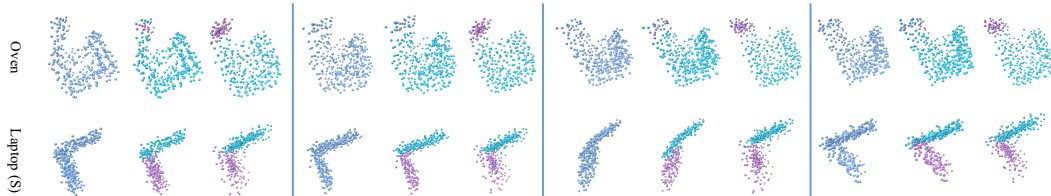

Figure 10: Visualization for the model performance on input data with random noise. Shapes for each three drawn from left to the right are input data corrupted by random normal noise, segmentation, and reconstruction, respectively. We align shapes here just for a better visualization, while they may be put into arbitrary poses for input.

Table 7: Comparison between the part pose estimation performance of different methods on all test categories (**partial point clouds**). "R" denotes rotation errors with the value format "Mean $R_{err}$/Median $R_{err}$". "T" denotes translation errors with the value format "Mean $T_{err}$/Median $T_{err}$". "J" denotes joint parameters estimation results with the value format "Mean $\theta_{err}$/Mean $d_{err}$". **ICP could not predict joint parameters.** Therefore, only the results of supervised NPCS and our method on joint prediction are presented. For all metrics, the smaller, the better. **Bold** numbers for best values, while *blue* values represent second best ones.

| | Method | Oven | Washing Machine | Eyeglasses | Laptop (S) | Safe | Laptop (R) | Drawer | Avg. |
|---|---|---|---|---|---|---|---|---|---|
| R | NPCS-EPN (supervised) | **2.51/2.27**, **2.93/2.64** | *4.71/3.84*, **8.56/7.46** | *7.26/6.08*, *23.39/17.33*, *20.86/18.76* | *21.40/23.56*, *29.90/32.71* | **6.64/5.76**, **5.43/5.19** | *9.39/8.75*, **6.75**/*6.14* | **21.74/10.80**, **22.92/10.18**, **25.10/14.16**, **7.34/6.83** | **13.34/10.73** |
| | Oracle ICP | 21.53/10.80, 20.68/20.50 | 32.42/17.82, 19.39/16.99 | 73.24/78.73, 68.74/74.09, 69.23/74.53 | 67.48/73.01, 63.22/68.23 | 38.72/34.44, 52.28/42.16 | 30.78/28.674, 42.06/39.25 | 82.93/82.64, 61.31/59.51, 54.39/52.82, 26.88/29.66 | 48.55/47.29 |
| | Ours | *11.77/7.87*, *10.83/9.15* | **1.61/1.52**, *12.81/12.51* | **4.69/3.77**, **9.56/5.36**, **7.53/6.12** | **10.18/5.30**, **11.10/5.22** | *15.38/14.46*, *21.91/19.04* | **8.50/6.85**, *6.92*/**5.66** | *2.60/1.79*, *2.60/1.79*, *2.06/1.79*, *2.06/1.79* | *8.36/6.47* |
| T | NPCS-EPN (supervised) | **0.028/0.030**, **0.028/0.023** | **0.034/0.030**, **0.033/0.028** | **0.085/0.075**, **0.056/0.052**, **0.057/0.049** | *0.263/0.253*, 0.286/0.236 | **0.022/0.021**, **0.034/0.034** | **0.048/0.043**, **0.047/0.044** | **0.441**/*0.365*, **0.367**/*0.343*, **0.549/0.299**, **0.081/0.065** | **0.145**/*0.117* |
| | Oracle ICP | 0.324/0.321, *0.169/0.171* | 0.322/0.311, *0.136/0.144* | *0.092/0.097*, 0.188/0.197, 0.185/0.193 | 0.265/0.278, *0.267/0.277* | 0.281/0.280, 0.246/0.248 | 0.280/0.289, 0.305/0.306 | *0.193/0.197*, *0.161/0.170*, *0.159*/**0.164**, *0.129*/0.132 | 0.218/0.222 |
| | Ours | *0.071/0.065*, 0.204/*0.120* | *0.179/0.164*, 0.253 /0.254 | 0.219/0.226, *0.169/0.166*, *0.177/0.171* | **0.044/0.034**, **0.031/0.025** | *0.030/0.030*, *0.100/0.104* | *0.088/0.082*, *0.070/0.067* | 0.046/0.046, 0.047/0.050, 0.122/0.131, 0.172/0.142 | *0.119*/**0.110** |
| J | NPCS-EPN (supervised) | 28.62/**0.092** | **8.05/0.194** | **20.11/0.221**, **20.11/0.239** | 10.91/0.155 | **11.23/0.084** | **12.25/0.134** | 11.21/- | **15.31/0.160** |
| | Ours | **5.24**/0.105 | 22.30/0.212 | 26.96/**0.087**, 26.96/0.260 | **10.83/0.142** | 55.16/0.170 | 18.02/0.170 | **7.43**/- | 21.61/0.164 |

Table 8: Comparison between the part segmentation performance of different methods (**partial point clouds**). The metric used for this task is Segmentation MIoU, calculated on 4096 points for each shape. Values presented in the table are scaled by 100. Larger values indicate better performance.

| | Oven | Washing Machine | Eyeglasses | Laptop (S) | Safe | Laptop (R) | Drawer |
|---|---|---|---|---|---|---|---|
| Oracle ICP | 75.83 | **73.07** | **68.92** | 54.01 | **66.90** | 59.96 | **58.38** |
| Ours | **87.07** | 51.73 | 56.80 | **84.94** | 44.64 | **86.04** | 45.45 |

Besides testing the performance of the proposed method on partial point clouds with occlusion patterns caused by viewpoint changes, we also test its effectiveness on noisy data. Specifically, we add noise for each point in the shape by sampling offsets for its x/y/z coordinates from normal distributions, *e.g.* $\Delta x \sim \mathcal{N}(0, \sigma^2)$, where we set $\sigma = 0.02$ here. Results on Oven and Laptop (S) are presented in Table 11. From the table, we can see the degenerated segmentation IoU on Oven's noisy data, while still relatively good part pose estimation performance. Another discovery is the even better joint axis orientation prediction, but larger offset prediction perhaps due to the poor segmentation. Besides, the shape reconstruction quality also drops a lot, probably due to the randomly shifted point coordinates. We can observe a similar phenomenon on Laptop (S). In Figure 10, we draw some examples for a qualitative understanding w.r.t. model's performance on noise data.

## B.7 VISUALIZATION OF PART-LEVEL EQUIVARIANT FEATURES

Aiming for an intuitive understanding w.r.t. the property output by the designed part-level equivariant network, we draw features output by the global equivariant network and part-level equivariant network for some laptop samples in Figure 11. From the figure, we can see that the point features of the non-motion part (base) do not change a lot when the moving part (display) rotates an angle. That echoes the wish for the part-level equivariance design to disentangle other parts' rigid transformation from the current part's feature learning.

Table 9: Comparison between the shape reconstruction performance of different methods (**partial point clouds**). The metric used in this task is unidirectional Chamfer L1 from the original input shape to the reconstructed shape. The smaller, the better.

| Method | Oven | Washing Machine | Eyeglasses | Laptop (S) | Safe | Laptop (R) | Drawer |
|---|---|---|---|---|---|---|---|
| EPN Li et al. (2021) | 0.040 | **0.043** | 0.044 | 0.032 | 0.020 | 0.026 | 0.079 |
| Ours | **0.035** | 0.062 | **0.041** | **0.025** | **0.019** | **0.024** | **0.061** |

Table 10: Comparison between the part pose estimation performance of different methods. Backbone used for NPCS is KPConv. "R" denotes rotation errors with the value format "Mean $R_{err}$/Median $R_{err}$". "T" denotes translation errors with the value format "Mean $T_{err}$/Median $T_{err}$". "J" denotes joint parameters estimation results with the value format "Mean $\theta_{err}$/Mean $d_{err}$". For all metrics, the smaller, the better. **Bold** numbers for best values.

| | Method | Oven | Washing Machine | Eyeglasses | Laptop (S) | Safe | Laptop (R) | Drawer | Avg. |
|---|---|---|---|---|---|---|---|---|---|
| R | NPCS-KPConv (supervised) | 44.16/43.09, 60.58/63.35 | 56.20/56.22, 50.16/51.38 | 51.99/53.97, 42.48/38.08, 42.29/38.11 | 55.67/66.44, 55.63/61.33 | 11.68/11.10, 43.48/42.22 | 49.98/68.43, 73.40/83.55 | 62.73/69.42, 56.16/60.34, 57.23/63.90, 48.76/46.82 | 50.74/53.99 |
| | Oracle EPN | **7.07/6.88**, 16.33/9.17 | 7.97/7.60, 33.56/20.49 | 54.01/13.09, 86.12/65.07, 116.56/119.23 | 18.33/9.73, 18.98/12.75 | 45.85/48.59, 38.03/27.67 | 20.46/14.03, 21.08/19.30 | 47.88/47.03, 30.84/25.23, 35.79/37.17, 43.83/39.46 | 37.81/30.73 |
| | Ours | 7.74/7.35, 4.07/3.97 | **7.49/7.37**, 19.27/19.19 | **8.16/8.21**, **12.29/10.89**, 12.53/9.88 | **7.34/5.16**, 10.41/9.34 | **9.03/9.09**, 13.83/13.59 | **5.71/3.61**, 3.64/2.84 | **3.18/2.73**, **3.18/2.73**, 3.18/2.71, 3.18/2.71 | **7.90/7.14** |
| T | NPCS-KPConv (supervised) | 0.133/0.121, 0.104/0.091 | 0.146/0.142, 0.066/0.065 | 0.401/0.326, 0.418/0.257, 0.396/0.263 | 0.233/0.203, 0.217/0.169 | **0.055/0.052**, 0.098/0.091 | 0.179/0.226, 0.161/0.174 | 0.791/0.742, 0.694/0.640, 1.005/0.942, 0.271/0.240 | 0.316/0.279 |
| | Oracle EPN | **0.031/0.030**, **0.058**/0.052 | **0.046/0.044**, 0.059/0.053 | 0.197/0.129, 0.128/0.118, 0.334/0.292 | 0.132/0.128, 0.117/0.090 | 0.157/0.157, 0.158/0.151 | **0.092/0.086**, **0.094/0.082** | 0.204/0.187, 0.177/0.166, 0.161/0.146, 0.290/0.282 | 0.143/0.129 |
| | Ours | 0.054/0.052, 0.067/**0.046** | **0.082/0.083**, 0.042/0.034 | **0.054/0.039**, **0.086/0.088**, 0.070/0.055 | **0.040/0.037**, 0.046/0.042 | 0.066/0.069, 0.037/0.035 | **0.021/0.019**, 0.027/0.026 | **0.096/0.096**, **0.097/0.092**, 0.108/0.105, 0.109/0.100 | **0.065/0.060** |
| J | NPCS-KPConv (supervised) | 55.62/0.194 | 55.01/0.149 | 60.58/0.329, 60.59/0.379 | 41.40/0.259 | 54.07/0.055 | 57.04/0.070 | 52.48/- | 54.60/0.205 |
| | Ours | **20.30/0.089** | **28.40/0.118** | **17.75/0.045**, 17.75/0.129 | 30.31/0.122 | 4.36/0.031 | 17.17/0.169 | **38.86/-** | **21.86/0.100** |

## B.8 EVALUATION STRATEGY FOR CATEGORY-LEVEL ARTICULATED OBJECT POSES

To evaluate the category-level part pose estimation performance of our model, we adopt the evaluation strategy used in Li et al. (2021).

For part-based metrics, we first feed a set of train shapes in the canonical articulation states and canonical object pose state to get a set of per-part pose predictions $\{P_i\}$. Then we can calculate the residual pose $\hat{P}_i$ for each part $i$ from the canonical part space defined by human to the canonical part space defined by the network from the pose prediction set (via RANSAC). After that, predicted pose from the canonical part space defined by human can be computed by applying the inverse residual pose estimation on the estimated per-part pose, *e.g.* $P_i \leftarrow \hat{P}_i^{-1} P_i$. When calculating the rotation and translation from part shape $X_1$ to $X_2$, we first centralize their bounding boxes ($\overline{X_1}$ and $\overline{X_2}$, respectively). Then, the transformation from $\overline{X_1}$ to $\overline{X_2}$ is taken as the transformation from $X_1$ to $X_2$.

For joint parameters, we take the angle error between the predicted joint axis orientation and the ground-truth axis orientation as the metric for joint axis orientation prediction. Metric for joint position prediction is set to the minimum line-to-line distance, following Li et al. (2020a). Only joint axis orientation prediction error is computed for prismatic joints.

## C DISCUSSION ON PART SYMMETRY

In this section, we discuss the part-symmetry-related problem that one would encounter in the part pose estimation problem. For rigid objects, the pose of a shape is ambiguous for symmetric shapes.

Table 11: Performance comparison of the proposed method on clean data and data corrupted by random normal noise.

| Category | Method | Seg. IoU | Mean $R_{err}(°)$ | Median $R_{err}(°)$ | Mean $T_{err}$ | Median $T_{err}$ | Joint Error | Chamfer L1 |
|---|---|---|---|---|---|---|---|---|
| Oven | Without noise | **76.22** | **7.74**, **4.07** | **7.35**, **3.97** | **0.054**, 0.067 | **0.052**, **0.046** | 20.30/**0.089** | **0.025** |
| | With noise | 55.35 | 9.84, 11.05 | 9.94, 9.99 | 0.073, **0.063** | 0.073, 0.057 | **9.28**/0.310 | 0.049 |
| Laptop (S) | Without noise | **82.97** | **7.34**, **10.41** | **5.16**, **9.34** | **0.040**, **0.046** | **0.037**, **0.042** | 30.31/0.122 | **0.024** |
| | With noise | 70.04 | 16.01, 13.27 | 11.47, 9.52 | 0.082, 0.067 | 0.075, 0.065 | 32.84/**0.029** | 0.044 |

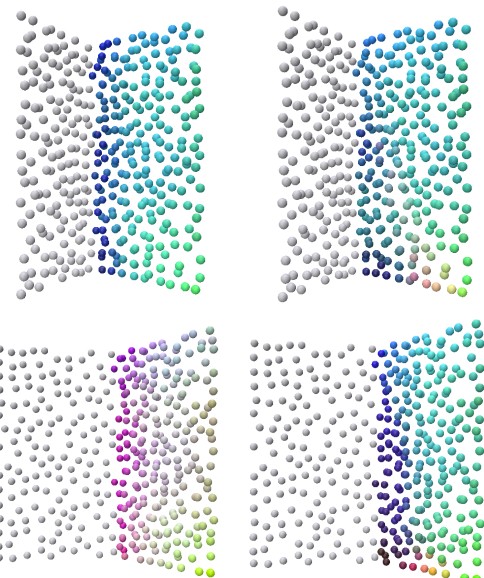

Figure 11: Visualization for an intuitive understanding w.r.t. the difference between the part-level equivariant feature and globa equivariant feature of a specific part. Visualized features are obtained by using the PCA algorithm to reduce the feature dimension to 3, which are further normalized to the range of [0, 1]. We only draw point features of the non-motion part with the moving part in gray. Features drawn on the left global equivariant features while those on the right are from the part-level equivariant network.

To say a shape $X$ is symmetric, we mean that there is a non-trivial SE(3) transformation $S_{A_0}$ such that $X = S_{A_0}[X]$. In those cases, the performance of the pose estimation algorithm may degenerate due to ambiguous poses. It is a reasonable phenomenon, however. But for articulated objects, we may have symmetric parts even if the whole shape is not a symmetric one. For those shapes, we still expect for accurate part pose estimation. It indicates that estimating part poses for each part individually is not reasonable due to part pose ambiguity. That's why we choose to model the relationship between parts, or specifically, the kinematic chain, joint parameters. Without such object-level inter-part modeling, we cannot get accurate part poses by estimating their pose individually, even using ground-truth segmentation. The comparison between Oracle EPN and our method in Table 10 can demonstrate this point to some extent.

