# OpenReview forum: "Self-Supervised Category-Level Articulated Object Pose Estimation with Part-Level SE(3) Equivariance"
_ICLR.cc/2023/Conference — ICLR 2023 poster_

### Official Review · Reviewer_GUmv · 2022-10-23

**Confidence:** 3
**Correctness:** 4
**Technical Novelty And Significance:** 4
**Empirical Novelty And Significance:** 4
**Recommendation:** 10

**Clarity, Quality, Novelty And Reproducibility:**

Clarity: Overall the writing is good, but I would suggest the authors revise more on the method section. Currently, it enumerates different components of the proposed system but lacks a good overview.

Quality: The paper is of high quality with solid methods and experiments.

Novelty: The proposed method is novel.

Reproducibility: The source code is provided. It should be fully reproducible.


**Strength And Weaknesses:**

Strength
1. It is very exciting to see that articulation emerges through the proposed self-supervised learning pipeline without any supervision. The system is complex, yet the results look very nice. Great work!
2. The paper proposed to integrate pose and part awareness in the equivariant point convolution, which naturally handles different articulated poses and unifies the point cloud feature in a canonical coordinate frame.
3. The results show that the proposed method achieves great performance, despite the system complexity and the simple self-supervised reconstruction loss.

Weakness
1. The paper is not well written, and the description of the proposed method is unclear. Different components are enumerated but a good overview of the method section is missing, for which I think Figure 3 in the appendix would be helpful. Specifically, what does “per-rotation articulated pose hypotheses” mean? What is the intuition behind the joint regularization loss?
2. It is not clear what role SE(3)-equivariant operations play in the proposed method. A baseline replacing all equivariant operations with non-equivariant operations is missing, or an argument on why equivariant is necessary is needed.
3. Experiments on real-world datasets are missing. (Figure 4 of ANCSH)


**Summary Of The Paper:**

The paper proposes to estimate articulated object pose on the category level in a self-supervised manner, by applying SE(3) equivariance on the part level. SE(3) equivariant features are extracted for each point from the input point cloud. The point cloud segmentation is conducted using slot attention, which divides the point cloud into articulated parts. Then the equivariant features of each segmented part are aggregated to estimate the part-level and object-level canonical space. Meanwhile, the kinematic chain and the joint parameters are regressed. The entire pipeline is trained using a shape reconstruction loss and a joint regularization loss, without manual annotation on object/part-level pose, kinematic structure, or joint parameters. Experiments show that the articulation structure and joint status emerge in such a self-supervised learning process and the estimation results are reported to be close to or better than the supervised counterpart.

**Summary Of The Review:**

The overall quality of the paper is very high. I’m excited to see decent results given the complex system and simple self-supervised loss. The description of the method still needs some improvement, but I would still argue for acceptance.

---

> ### Author Response · Authors · 2022-11-13
> **Response to Reviewer GUmv (Part 2 of 2)**
>
> 2. **About the role of SE(3)-equivariant operations in the proposed method.**
>
>    - **Explanations on why SE(3)-equivariance is needed:** Estimating articulated object pose requires us to disentangle pose-related information and pose-invariant information from input shapes without annotations.
>
>      It is a highly ill-posed problem in nature. SE(3) equivariant features are then important for us to solve the task for the following reasons:
>
>      - Using SE(3) invariant features spares the efforts of the network to predict *invariant* canonical part shapes.
>      - Predicting *residual* pose of a small quantity from per-rotation features reduces the difficulty of pose estimation.
>
>      By contrast, non-equivariant features require the network to take a large effort for such disentanglement, resulting in very poor performance in the self-supervised setting.
>
>    - **Empirical evidence:** In the submission, we validate the necessity of SE(3) equivariant features to solve the self-supervised task from the comparison between **NPCS-EPN** in the main paper (Table 1) and **NPCS-KPConv** in the Appendix (Table 10).
>
>      - For **NPCS-EPN,** we change its original backbone (PointNet++) to EPN. NPCS-EPN then leverages equivariant features for invariant shape prediction in the normalized part coordinate space.
>
>      - As shown in the following table, ablating SE(3) equivariant features would lead to a huge performance drop even for *the supervised method*.
>
>        |                              | Mean $R_{err}$ | Median $R_{err}$ | Mean $T_{err}$ | Median $T_{err}$ | Joint          |
>        | ---------------------------- | -------------- | ---------------- | -------------- | ---------------- | -------------- |
>        | **NPCS-KPConv (supervised)** | 50.74          | 53.99            | 0.316          | 0.279            | 54.60/0.205    |
>        | **NPCS-EPN (supervised)**    | **5.38**       | **4.70**         | **0.023**      | **0.019**        | **9.27/0.089** |
>
>      - We did not report the performance of our method without equivariance in the submission due to its extremely poor performance even with degenerated segmentations sometimes.
>
>        We are happy to provide the results if needed.
>
> 3. “**Experiments on real-world datasets are missing. (Figure 4 of ANCSH)”**
>
>    - **Experiments on real-world datasets:** In our submission, we do test the method on a real-world category-level dataset HOI4D [1], much more challenging that the instance-level dataset [2] used in Figure 4 of ANSCH.
>    - We do not use exactly the same real dataset [2] as ANSCH since that dataset only contains four different instances with each corresponding to one different category. ANSCH assumes very strong supervision so it could degenerate into an instance-level pose estimation method when trained and evaluated on [2]. This dataset does not support our self-supervised category-level training though since we need shape variations to facilitate category-level shape alignment.
>    - We have experimented on two articulated object categories namely Safe and Laptop (R) from HOI4D [1] as shown in Table 1 and Table 7 in the main paper. It can be seen that our method performs very well. We would be happy to also include a visualization like the Figure 4 of ANCSH if needed.
>
>
>
> Thanks again for your careful review and recognization of our work. We sincerely hope the above explanations could address your concerns about the work. We are happy to take any further questions.
>
>
>
> **References**
>
> *[1] Liu, Y., Liu, Y., Jiang, C., Lyu, K., Wan, W., Shen, H., ... & Yi, L. (2022). HOI4D: A 4D Egocentric Dataset for Category-Level Human-Object Interaction. In Proceedings of the IEEE/CVF Conference on Computer Vision and Pattern Recognition (pp. 21013-21022).*
>
> *[2] Frank Michel, Alexander Krull, Eric Brachmann, Michael Ying Yang, Stefan Gumhold, and Carsten Rother. Pose estimation of kinematic chain instances via object coordinate regression. In BMVC, pages 181–1, 2015. 3, 6, 8*

---

> > ### Comment · Reviewer_GUmv · 2022-12-01
> > **Thanks for the response**
> >
> > Thanks for the response and they addressed my concerns. I would like to keep my initial rating and appreciate the great work!

---

> > > ### Author Response · Authors · 2022-12-06
> > > **Thanks a lot**
> > >
> > >
> > > Thank you for your support!
> > >
> > > Best regards,
> > >
> > > Authors

---

> ### Author Response · Authors · 2022-11-13
> **Response to Reviewer GUmv (Part 1 of 2)**
>
>
>
> Dear Reviewer GUmv,
>
>
>
> Thanks for your detailed and valuable feedback.
>
> Below we address your concerns.
>
>
>
> 1. **On the presentation of the proposed method**
>
>    > The paper is not well written, and the description of the proposed method is unclear. Different components are enumerated but a good overview of the method section is missing, **for which I think Figure 3 in the appendix would be helpful**. Specifically, what does “per-rotation articulated pose hypotheses” mean? What is the intuition behind the joint regularization loss?
>
>    - Thanks for your careful reading and all your suggestions. We have revised Section 3 and Figure 1 in our revison to improve our presentation on the method section.
>
>    - **Method overview:** We introduce the design of our method in the first paragraph of the method section, including why we pursue part-level SE(3) equivariance, how we factorize and compose information from input shapes, and the self-supervised reconstruction task.
>
>      We have reorganized the first paragraph of the Section 3.2 in the view of shape reconstruction. We hope such two overview paragraphs could make the big picture of the method well delivered to readers.
>
>    - **Why we choose to use Figure 1 in the main text:** Due to the sophisticated disentanglement strategy, we choose to only draw an abstracted pipeline in Figure 1, leaving details to the text of Section 3.
>
>    - We have drawn **a separate figure** to further explain the disentanglement and compositional reconstruction design on the right of the abstracted pipeline. We hope they together could make both the high-level information and crucial details much more clear.
>
>    - **Clarifications on some details:**
>
>      - **per-rotation articulated pose hypotheses**: the concept of “per-rotation” is based on the format of our equivariant features (per-point per-rotation features). Such features, denoted as $\mathbf{F}\in \mathbb{R}^{N \times K \times \vert G\vert }$, contain one feature vector $\mathbf{F}[g]\in \mathbb{R}^{N \times K }$ for each major rotation element $g\in G$. We predict a residual pose hypothesis for each rotation element $g\in G$ for pose estimation. That’s what **“per-rotation articulated pose hypotheses”** mean.
>      - **the intuition behind the joint regularization loss**: The design intuition is enforcing assembled parts to be connected by their predicted joint. We then use *a shifted pivot point set* to simulate joints. By minimizing its distance to part point clouds before and after articulated motion, we could let the network predict a “real” joint.

---

### Official Review · Reviewer_QLf2 · 2022-10-25

**Confidence:** 2
**Correctness:** 4
**Technical Novelty And Significance:** 3
**Empirical Novelty And Significance:** 2
**Recommendation:** 6

**Clarity, Quality, Novelty And Reproducibility:**

This paper is well written and organized with substantial experiments, clear contributions and high quality. The problem definition is clear, and the proposed method has distinct details. Also, the paper proposed approach seems to be novel and useful.

**Strength And Weaknesses:**

Strength:
1. The paper considers a challenging setting, where the human labels of the articulated object pose estimation task is not given. Category-level articulated object pose estimation is a very popular task. Getting more insight about each different method can be valuable to many people.
2. The proposed method is novel and effective.
3. The paper is well written and organized.
4. Comprehensive experiments are conducted to support the effectiveness of the proposed method.


**Summary Of The Paper:**

In this paper, the authors present a novel self-supervised strategy to reduce the heavy annotations needed for supervised learning methods for Category-level articulated object pose estimation. Specifically, they design a pose-aware equivariant point convolution operator to learn part-level SE(3)-equivariant features. Then, they propose a self-supervised framework to achieve the disentanglement of canonical shape, object structure, and articulated object poses. After that, the authors further predict articulated object poses as per-part rigid transformations describing how parts transform from their canonical part spaces to the camera space. Extensive experiments demonstrate the effectiveness of their method.

**Summary Of The Review:**

The authors study a challenging problem and propose a novel approach. This paper is well-organized and easy to follow. In addition, the experimental results are promising.

---

> ### Author Response · Authors · 2022-11-13
> **Response to Reviewer QLf2**
>
> Dear Reviewer QLf2,
>
> Thanks for your detailed reviews and the recognization of our work.
>
> Since we do not see your concerns about this paper from the review, so please don’t hesitate to let us know if you have any questions or any clarifications that we can offer.
>
> We are very willing to address any of your concerns about the work.
>
> Thanks again for your careful review. We are happy to take any questions.

---

> ### Author Response · Authors · 2022-12-06
> **Looking forward to your feedback**
>
> Dear Reviewer QLf2,
>
> Thank you again for your time.  As the deadline for discussion is approaching, we do wish to hear from you to see if you have any concerns about the work.
>
> We have followed suggestions from other reviewers to improve the presentation of Section 3.  We also gave detailed answers to their questions, where common ones are like how the number of parts is determined and how our strategy can support us to predict intra-category aligned canonical part shapes.
>
> We appreciate your recognition of our work's challenging setting and effectiveness.  Please do not hesitate to let us know if you have any questions.  Thanks!
>
> Best regards,
>
> Authors

---

### Official Review · Reviewer_gB2X · 2022-10-25

**Confidence:** 2
**Correctness:** 4
**Technical Novelty And Significance:** 3
**Empirical Novelty And Significance:** 2
**Recommendation:** 6

**Clarity, Quality, Novelty And Reproducibility:**

The overall quality of the work is good. Although the originality is arguable (the overall framework has been proposed on object-level), the progress made by this paper is interesting to the community.

**Strength And Weaknesses:**

Strengths:
- The problem being studied is important and general enough to be applied in other scenarios.
- The proposed method is intuitive in the sense that we as humans would perceive the canonical shapes of the object parts, not just the entire object.

Weakness:
- The presentation could be slightly improved. For example, some information in Figure 3 could be put into Figure 1 to help readers get a clearer picture.
- From my understanding of the paper, the canonical shapes Z_i are instance-level concepts. Should we have a loss to make sure the shapes are aligned between different instance objects of the same category?
- How are the number of parts controlled in the algorithm?
- What is the stopping criterion for the iterations?

**Summary Of The Paper:**

This paper tackles the problem of category-level articulated object pose estimation, which requires articulated pose estimation of an unseen object from a known category. The authors propose a self-supervised method to reduce the need for annotations. This method factorizes the input point clouds into canonical shapes and part-level poses. By designing a network that learns a part-level equivariance property, the object shape could be reconstructed from the canonical parts. Experiments show that the proposed method can perform on par with or even better than existing methods that are supervised.


**Summary Of The Review:**

Overall I am leaning towards accepting the paper, and I look forward to the authors' response to the above questions.

---

> ### Author Response · Authors · 2022-11-13
> **Response to Reviewer gB2X (Part 2 of 2)**
>
> 3. **“How are the number of parts controlled in the algorithm?”**
>
>    - The number of parts depends on how many **modes of rigid part motions** could be discovered from a category of shapes so that we can achieve the minimum reconstruction error.
>    - It is determined by the network through our category-level training.
>    - Note that the number of slots (K) for grouping points *only serves as the upper bound* of the number of parts.
>
> 4. **The stopping criterion for iterations**
>
>    - The iteration would stop if any of the following two conditions are satisfied:
>      - Get similar segmentation results in two adjacent iterations.
>        - For two hard segmentation matrices at the $t$-th iteration and the previous iteration $\mathbf{W}^t, \mathbf{W}^{t-1} \in \mathbb{R}^{N\times K}$ with 0/1 values, K is the number of slots, we consider them similar if $\sum \vert \mathbf{W}^t - \mathbf{W}^{t-1}\vert \le \delta \cdot N$ , where $\delta = 0.1$.
>        - It means that if the point-part assignment is the same for not less than 95% of points in two adjacent iterations, the iteration could be stopped.
>      - Reach the maximum number of iterations, which is set as a hyper-parameter, *i.e.,* 5 in our experiments.
>
> 5. **On the originality of our work**
>
>    - Further, we would like to clarify two main originalities of our work:
>
>      - We propose a **pose-aware point convolution module (Section 3.1)**. Based on that, we design our **part-level SE(3) equivariant network**. To our best knowledge, it is **the first work** that tries to pursue part-level SE(3) equivariance.
>
>      - We propose a **disentanglement strategy for articulated object pose estimation**, where how to predict part-level properties, how to model inter-part relationship, and how to recover the articulation chain are carefully designed.
>
>        Ignoring the part-object relationship would result in severe part symmetry-related issues and therefore poor pose estimation performance even with ground-truth segmentations as discussed in section B.5 and section C.
>
>
>
> Thanks again for your careful and detailed reviews! We sincerely hope the above explanations could address your concerns. We are happy to take any further questions.
>
>
>
> **References**
>
> *[1] Li, X., Wang, H., Yi, L., Guibas, L. J., Abbott, A. L., & Song, S. (2020). Category-level articulated object pose estimation. In Proceedings of the IEEE/CVF conference on computer vision and pattern recognition (pp. 3706-3715).*
>
> *[2] Mehr, E., Lieutier, A., Bermudez, F. S., Guitteny, V., Thome, N., & Cord, M. (2018). Manifold learning in quotient spaces. In Proceedings of the IEEE Conference on Computer Vision and Pattern Recognition (pp. 9165-9174).*
>
> *[3] Li, X., Weng, Y., Yi, L., Guibas, L. J., Abbott, A., Song, S., & Wang, H. (2021). Leveraging SE (3) Equivariance for Self-supervised Category-Level Object Pose Estimation from Point Clouds. Advances in Neural Information Processing Systems, 34, 15370-15381.*
>
> *[4] Locatello, F., Weissenborn, D., Unterthiner, T., Mahendran, A., Heigold, G., Uszkoreit, J., ... & Kipf, T. (2020). Object-centric learning with slot attention. Advances in Neural Information Processing Systems, 33, 11525-11538.*

---

> ### Author Response · Authors · 2022-11-13
> **Response to Reviewer gB2X (Part 1 of 2)**
>
> Dear Reviewer gB2X,
>
>
>
> Thanks for your valuable feedback and insightful suggestions.
>
> Below we address your concerns.
>
>
>
> 1. **“The presentation could be slightly improved. For example, some information in Figure 3 could be put into Figure 1 to help readers get a clearer picture.”**
>
>    - Thanks for your suggestions! Due to the sophisticated compositional reconstruction process in our method, we choose to only draw an abstracted pipeline to explain our high-level idea in the main text.
>    - We do appricate your suggestions and **have added a separate figure** on the right side of the abstracted pipeline to further illustrate the compositional reconstruction process. We hope them together could make both the big picture and some crucial details of the method clearly delivered to readers.
>
> 2. **“Should we have a loss to make sure $Z_i$ are aligned between different instance objects of the same category?”**
>
>    - **Why we can predict cross-instance aligned $Z_i$ with no extra losses:** We indeed do not explicitly enforce cross-shape alignment in the canonical shape reconstruction. However, we find the network aligns parts across all instances automatically for reconstruction. It is due to the following two reasons:
>
>      - Pose-invariant features for reconstruction. Such features only encode geometry information and are **similar** across part shapes from the same category.
>      - Taking similar geometric features, the network **automatically chooses to reconstruct canonical part shapes in the aligned pose**. It is because of networks’ **“lazy”** nature since such reconstructions are the easiest thing to conduct, as also observed in [2,3].
>
>      Therefore, the category-level aligned reconstructions could emerge automatically during our category-level training.
>
>      - **Experimental observations:** In practice, we can observe good cross-instance alignment from the reconstructed canonical part shapes, *e.g.,* the rightmost ones in every three shapes drawn in Figure 2, 7, 8.
>
>    - **Why we do not add a loss:** Adding a loss for aligned canonical shape reconstruction is a great idea and we do appreciate it. In our design, we choose no explicit losses for the following two considerations:
>
>      - Adding a loss may make the self-supervised optimization hard to solve.
>      - Our empirical results demonstrate the effectiveness of the design without an additional loss.

---

> ### Author Response · Authors · 2022-12-06
> **Looking forward to your feedback**
>
> Dear Reviewer gB2X,
>
> Thank you again for your time. As the deadline for discussion is approaching, we do wish to hear from you to see if our response resolves your concerns. We are happy to provide any additional clarifications if needed.
>
> We have followed your suggestions to improve Figure 1. In the response, we explained why we choose to add no extra loss when predicting $Z_i$ but can still get intra-category aligned predictions, how the number of parts is determined, and the stopping criterion for iterations. We also further clarified the **main originalities** of our work.
>
> We would appreciate you kindly checking our response. Please do not hesitate to let us known if you have any further questions. Thanks!
>
> Best regards,
>
> Authors

---

### Official Review · Reviewer_sxak · 2022-10-26

**Confidence:** 3
**Correctness:** 4
**Technical Novelty And Significance:** 2
**Empirical Novelty And Significance:** 3
**Recommendation:** 6

**Clarity, Quality, Novelty And Reproducibility:**

The overall idea is clear. But the description in Section 3 is a bit cluttered. It may be helpful to re-organize a bit to make it easier to read. For example, introduce the modules doing canonical shape estimation first, followed by the modules that align them to the observed point cloud. For each module. state clearly which point cloud it is trying to be aligned with, the canonical one or the observed one.
The work is of good quality. Code is provided so reproducibility looks good.

**Strength And Weaknesses:**

### Strengths
- Self-supervision is a good idea when it comes to articulated and part-level tasks, where the annotation effort is high.
- Many biases are carefully engineered to make sure the correct feature (invariant/equivariant) is used.
- Based on the experimental results, the proposed self-supervision pipeline gets reasonable preformance.

### Weaknesses / Questions
- Section 3 is wordy and it would be helpful to re-organize.
- In Part-assembling parameters prediction, only the translation is predicted. Is translation alone sufficient? It seems that this will make the reconstruction of canonical shape harder.
- 'we first construct an adjacency confidence graph from object parts and then extract its maximum spanning tree consisting of the set of confident adjacency edges' How are the edge weights determined? How do you define the subset that is ‘the set of confident adjacency edges’?
- 'The part proposal module groups N points in the input shape X into K parts' How is K determined? Does that affect the final performance?
- Since the part segmentation learned by the proposed method comes from self-supervision, how can it be directly compared to the ICP and NPCS where there is ground truth part-segmentation?


**Summary Of The Paper:**

This paper is working on self-supervised articulated object pose estimation at a category level. The objective is to estimate the canonical shape, articulated structure, and transformation that assembles all canonical shapes together. The supervision comes from the overall reconstruction. Experiments are carried out on 7 categories from 3 datasets. Result shows that the proposed method outperforms or on par with baselines, some of which have external supervision.

**Summary Of The Review:**

The proposed method alleviates the need of data annotation for articulated object pose estimation. The result is fair given that it's self-supervised.

---

> ### Author Response · Authors · 2022-11-13
> **Response to  Reviewer sxak (Part 2 of 2)**
>
>
>
> 3. **Kinematic chain prediction: Edge weights prediction and how to define the set of confident adjacency edges**
>
>    - **Edge weights** are predicted from invariant features (extracted by an SO(3)-PointNet from part equivariant features) of each part pair by an MLP module.
>    - **The set of confident adjacency edges** is found by finding the [maximum spanning tree](https://mathworld.wolfram.com/MaximumSpanningTree.html) from the fully-connected graph with parts as its nodes and predicted pair-wise confidence values as its edge weights.
>    - **Supervision for edge weights prediction:** The edge weights prediction is supervised jointly with joint constraints. Specifically, the chamfer distance between the pivot point set and the transformed segment point cloud set is further multiplied by the predicted confidence value. Intuitively, a large distance indicates a poor adjacency prediction, so the confidence should be penalized.
>
> 4. **How is the number of parts K determined? Does that affect the final performance?**
>
>    - K in the part proposal module is set as a hyper-parameter (*i.e.,* 4 in our experiments).
>    - Despite this, K only determines the upper limit of the number of parts. The number of parts in the articulated object is determined by the network automatically.
>    - Specifically, it depends on how many modes of rigid part motions could be discovered from a category of shapes so that we can achieve the minimum reconstruction error.
>    - **Sensitivity to K:** Since the number of parts is determined by how many rigid part motion modes could be discovered from the shape category, our method is not sensitive to K. For instance, In our experiments, the performance varies only slightly when K is set from 4 to 9.
>
> 5. **How can our method, learning segmentations via a self-supervised task, be directly compared to the ICP and NPCS where there is ground truth part-segmentation?**
>
>    - **About the evaluation:** For each method, we compare the predicted values to corresponding ground-truth values calculated using ground-truth segmentations. The results could therefore be compared across different methods for performance evaluation.
>
>    - Indeed, comparing to ICP and NPCS with ground-truth segmentations either during the test time or the training time is **not fair to our method**. We compare them due to the lack of previous works with exactly the same setting as ours.
>
>      Even so, we can achieve **comparable or even better performance**, demonstrating the effectiveness of our self-supervised strategy convincingly.
>
>
>
> Thanks again for your careful and valuable review! We sincerely hope the above explanations could address your concerns. We are happy to take any further questions.
>
>
>
> **References**
>
> *[1] Locatello, F., Weissenborn, D., Unterthiner, T., Mahendran, A., Heigold, G., Uszkoreit, J., ... & Kipf, T. (2020). Object-centric learning with slot attention. Advances in Neural Information Processing Systems, 33, 11525-11538.*

---

> ### Author Response · Authors · 2022-11-13
> **Response to  Reviewer sxak (Part 1 of 2)**
>
> Dear Reviewer sxak,
>
>
>
> Thanks for your detailed reviews and constructive suggestions.
>
> Below we address your questions.
>
>
>
> 1. **”Section 3 is wordy and it would be helpful to re-organize”**
>
>    > For example, introduce the modules doing canonical shape estimation first, followed by the modules that align them to the observed point cloud. For each module. state clearly which point cloud it is trying to be aligned with, the canonical one or the observed one.
>
>    - Thanks for your suggestions for Section 3.
>    - In the submitted version, we organize section 3 from the view of shape composition workflow by
>      - First introducing how we group points into parts;
>      - Then explaining the *canonical part shape prediction* module;
>      - Followed by how we predict different kinds of pose-related parameters to *assemble parts for the observed point cloud*.
>      We think it is similar to your proposed approach in nature. For instance, it is clear w.r.t. **canonical shape estimation** and how to **align them to the observed point cloud.**
>    - **We have improved Section 3 in our revision. And we highlight two modifications as follows:** 1) We clarify the objective of each prediction step, *e.g.,* which point cloud to align with; 2) We reorganize the overview paragraph of Section 3.2 from the view of shape reconstruction. All of the above modifications are highlighted in blue.
>
>
>
>
>
> 2. **About using translations alone as part-assembling parameters**
>
>    - **"Is translation alone sufficient?"** Yes, predicting translation alone is sufficient for us, which could be demonstrated by our experiments:
>      - **Quantitative experimental evidence:** The performance of category-level articulated object pose estimation relies heavily on and could therefore indicate the quality of category-level canonical shape reconstructions. As shown in our experimental results in Table 1 and Tale 7, our canonical shape reconstruction **is sufficient to support articulated object pose estimation.**
>      - **Qualitative experimental evidence:** We could also observe high-quality point clouds and **good cross-instance alignment** from the canonical shape reconstructions shown in Figure 2, 7, and 8.
>    - **“It seems that this will make the reconstruction of canonical shape harder.“**
>      - **Requirements for canonical part shape reconstructions:** Predicting translations alone requires **compatible** canonical part shape reconstructions. For instance, two adjacent parts should be seamed together by their joint in the canonical object space.
>      - **Two designs in our method could encourage such compatible canonical part shape prediction:** 1) Our reconstruction process which requires the reconstructed canonical object shape to support correct articulation motion. 2) Our joint regularization which encourages the connectivity between part shapes both before and after articulation.
>      - **Practical observations:** Though it seems to make the reconstruction harder due to the restricted reconstruction freedom, we do not observe that affects our reconstruction quality or the phenomena where networks struggle to predict cross-instance aligned canonical part shapes. **We can solve the problem well with our designs.**
>      - **An early trial on rigid transformation for part-assembling parameterization:** We tried to use rigid transformations as part-assembling parameters in our early experiments. We observe this strategy together with the following articulated object pose estimation design makes the problem hard for networks to solve probably due to the *heavy optimization burden.*
>      - **As a further comment:** Restricting canonical part shape prediction freedom is necessary if we want to build relationships between different parts for articulated object pose estimation. Predicting compatible canonical part shapes is a feasible strategy in our practice.

---

> ### Author Response · Authors · 2022-12-06
> **Looking forward to your feedback**
>
> Dear Reviewer sxak,
>
> Thank you again for your time. As the deadline for discussion is approaching, we do wish to hear from you to see if our response resolves your concerns. We are happy to provide any additional clarifications if needed.
>
> We have followed your suggestions to improve the presentation of Section 3. In the response, we explained how the number of parts is determined, why we only predict translations as part-assembling parameters and whether it is sufficient, and how the set of confident adjacency edges is determined, etc.
>
> We would appreciate you kindly checking our response. Please do not hesitate to let us known if you have any further questions. Thanks!
>
> Best regards,
>
> Authors

---

### Author Response · Authors · 2022-11-22
**Thanks for all your comments and look forward to post-rebuttal feedbacks**

Dear Reviewers:

Thanks again for all of your detailed reviews and constructive suggestions, which have helped us improve our paper a lot.

We have given our responses to all of your questions and have incorporated your suggestions in the revised paper (highlighted in blue). We hope our explanations and revisions could address your concerns.

Please don’t hesitate to let us know if you have any further questions. We would love to convince you of the merits of the paper.  We appreciate your suggestions.

Thanks!

---

### Decision · Program_Chairs · 2023-01-20

**Decision:**

Accept: poster

**Justification For Why Not Higher Score:**

- The novelty of the proposed work is limited, as it extends the self-supervised learning framework for object-level pose estimation framework from [Li et al. 20] to part-level.
- Not sure whether this framework will be of interest to a large audience at ICLR, since it is specifically tailored for object pose estimation problem.


**Justification For Why Not Lower Score:**

All reviewers voted to accept the paper as they found the idea as intuitive and the performance impressive.

**Metareview: Summary, Strengths And Weaknesses:**

The paper proposes a self-supervied learning method for object pose estimation, for reduced annotation cost. Specifically, the proposed self-supervised learning framework estimates the canonical shape, articulated structure, and transformation for assembling all canonical shapes together, with a pose-aware equivariant point convolution operator to learn part-level SE(3)-equivariant features. The experimental validation of the proposed method on benchmark datasets show that it performs on part with, or outperforms baselines, even including supervised methods.

All reviewers leaned toward acceptance, as they found the idea of learning part-level equivariance reasonable as well as intuitive, the method practical and effective as it achieves impressive performance without any labels.

However, there was some concern on the presentation and writing as the paper does not provide a good overview of the method, and does not clearly describe what role SE(3)-equivariant operations play. Also, there was a concern on the limited novelty of the proposed work over [Li et al. 21], which proposed a similar framework for object-level pose estimation.

I also agree with the reviewers that the contribution of the paper is rather limited as they extend the object-level pose estimation framework in [Li et al. 21] to part-level, and that the presentation needs to be improved. Specifically, the authors use the terminology “SE(3)-equivariance “ without even introducing its concept. However I recommend a “weak accept” since the work is well-received by the researchers working on the topic, and may have some practical impact due to its good performance.

[Li et al. 21] Leveraging SE(3) Equivariance for Self-Supervised Category-Level Object Pose Estimation, NeurIPS 2021


**Note From Pc:**

if the above contains the word "oral" or "spotlight" please see: "oral" presentation means -> notable-top-5% and "spotlight" means -> notable-top-25%. As stated in our emails, we are disassociating presentation type from AC recommendations